# TTOM: Test-Time Optimization and Memorization for Compositional Video Generation

**Leigang Qu**[1*]    **Ziyang Wang**[1*]    **Na Zheng**[1]    **Wenjie Wang**[2†]
**Liqiang Nie**[3]    **Tat-Seng Chua**[1]
[1]National University of Singapore    [2]University of Science and Technology of China
[3]Harbin Institute of Technology (Shenzhen)
{leigangqu, wenjiewang96, zhengnagrape, nieliqiang}@gmail.com
wangziyang@u.nus.edu    dcscts@nus.edu.sg
https://ttom-t2v.github.io/

## ABSTRACT

Video Foundation Models (VFMs) exhibit remarkable visual generation performance, but struggle in compositional scenarios (*e.g.*, motion, numeracy, and spatial relation). In this work, we introduce **Test-Time Optimization and Memorization (TTOM)**, a model-agnostic framework that aligns VFM outputs with spatiotemporal layouts during inference for better text-image alignment. Rather than direct intervention to latents or attention per-sample in existing work, we integrate and optimize new parameters guided by a general layout-attention objective. Furthermore, we formulate video generation within a streaming setting, and maintain historical optimization contexts with a parametric memory mechanism that supports flexible operations, such as insert, read, update, and delete. Notably, we found that TTOM disentangles compositional world knowledge, showing powerful transferability and generalization. Experimental results on the T2V-CompBench and Vbench benchmarks establish TTOM as an effective, practical, scalable, and efficient framework to achieve cross-modal alignment for compositional video generation on the fly.

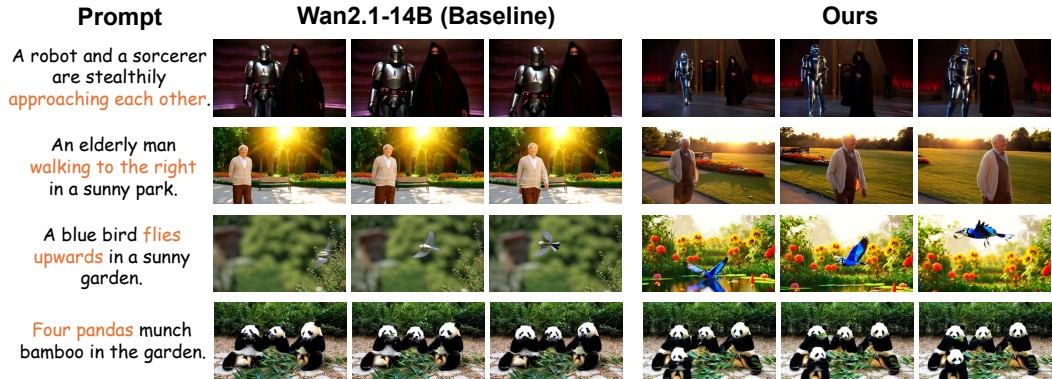

Figure 1: Current video generative models (Wan et al., 2025) still suffer from text-video misalignment problems in compositional scenarios. We introduce a test-time optimization and memorization method that substantially enhances alignment while maintaining high visual fidelity.

## 1 INTRODUCTION

Recent years have witnessed a rapid stride in video generation (Ho et al., 2022; Polyak et al., 2024) and world simulation (Brooks et al., 2024). Benefiting from the significant advance in flow matching (Lipman et al., 2022; Esser et al., 2024) and diffusion transformers (DiTs) (Peebles & Xie,

---

*Equal contribution.

†Corresponding author.

2023), current text-to-video generation (T2V) models (Yang et al., 2024b; Kong et al., 2024) are able to synthesize realistic and vivid videos. However, even state-of-the-art models still suffer from *text-video misalignment* in compositional scenarios, *i.e.*, composing multiple objects, attributes, and relations into a complex scene (Sun et al., 2025; Huang et al., 2024).

Remarkable efforts have been made to improve text-video alignment by introducing explicit spatiotemporal layout guidance (Lian et al., 2023b; He et al., 2025) as a bridge between text prompts and videos. In this paradigm, a layout represented by bounding box (bbox) sequences—each corresponding to an object—is first generated by a large language model (LLM), and subsequently used to optimize intermediate representations (*e.g.*, latents and attention maps) by enforcing alignment with the layout constraints. Despite notable progress, this line of work faces several limitations: 1) direct intervention on intermediate representations may disrupt feature distributions, leading to degraded video quality (*e.g.*, inconsistency, flickering artifacts, or even collapse); 2) the per-sample control paradigm treats each case independently, thereby neglecting the *historical context* of previously generated videos; and 3) these methods do not enhance the intrinsic capability of generative models, since interventions on one sample fail to generalize to others.

In real-world scenarios, models are not presented with a set of independent test cases but with a continuous stream of user prompts, as illustrated in Fig. 2. Previous successful generations can serve as valuable references for future cases. In light of this, our core idea is to *formulate compositional video generation in a streaming setting, leveraging history test-time optimization as context for future inference*. However, modeling history contexts is non-trivial, as it requires addressing the fundamental challenge of how to represent, store, reuse, and update past information.

In this work, we introduce Test-Time Optimization and Memorization (**TTOM**) for compositional T2V. For the first test case (cold start), a large language model (LLM) first derives a spatiotemporal layout from the text prompt, serving as a controllable condition. Conditioned on this layout, we perform Test-Time Optimization (TTO) by instantiating and updating sample-specific parameters, thereby steering video generation toward adherence to the prescribed layout. Through this process, the compositional patterns embedded in the layout, such as motion, numeracy, and multi-object interactions, are attained and saved into the new parameters. After optimization, the parameters are unloaded and stored in memory, using the extracted layout-related keywords from the prompt as keys. For subsequent test cases, when matched parameters are retrieved, they can be integrated into the foundation model and further optimized for improved adaptation, or directly applied for efficient inference. If no match is found, new parameters are initialized, optimized, and subsequently recorded in memory to expand the memory. In practice, the memory is assigned a fixed capacity; when full, items are removed according to predefined policies (*e.g.*, least frequently used). This mechanism enables the flexible reuse of informative optimization results by preserving historical context, thereby enhancing both efficiency and scalability. In summary, our main contributions are as follows.

- We propose a test-time optimization framework without any supervision for compositional T2V. With spatiotemporal layout as guidance, we incorporate and optimize lightweight parameters for each data sample.

- We present a parametric memory mechanism to maintain historical optimization context, which naturally supports lifelong learning. A series of operations (*e.g.*, insert, read, update, and delete) for memory ensures flexibility, scalability, and efficiency.

- Extensive experimental on two T2V benchmarks (Sun et al., 2025; Huang et al., 2024) demonstrate the effectiveness and superiority of the proposed method. Notably, it achieves relative improvements of 34% and 14% over CogVideoX-5B (Yang et al., 2024b) and Wan2.1-14B (Wan et al., 2025) on T2V-CompBench (Sun et al., 2025), respectively.

## 2 RELATED WORK

**Compositional Visual Generation**. Despite the thrilling progress in foundation models (Yang et al., 2024b; Kong et al., 2024; Polyak et al., 2024; Seawead et al., 2025), compositional generation (Chefer et al., 2023; Feng et al., 2022; Qu et al., 2024b; 2025b) remains an open challenge and has attracted substantial research interest. To enhance text-image alignment in compositional scenes, a rich line of studies (Qu et al., 2023; Feng et al., 2023) explores incentivizing the layout planning ability of LLMs (Brown et al., 2020; Ouyang et al., 2022) to enable layout-guided controllable generation (Tian

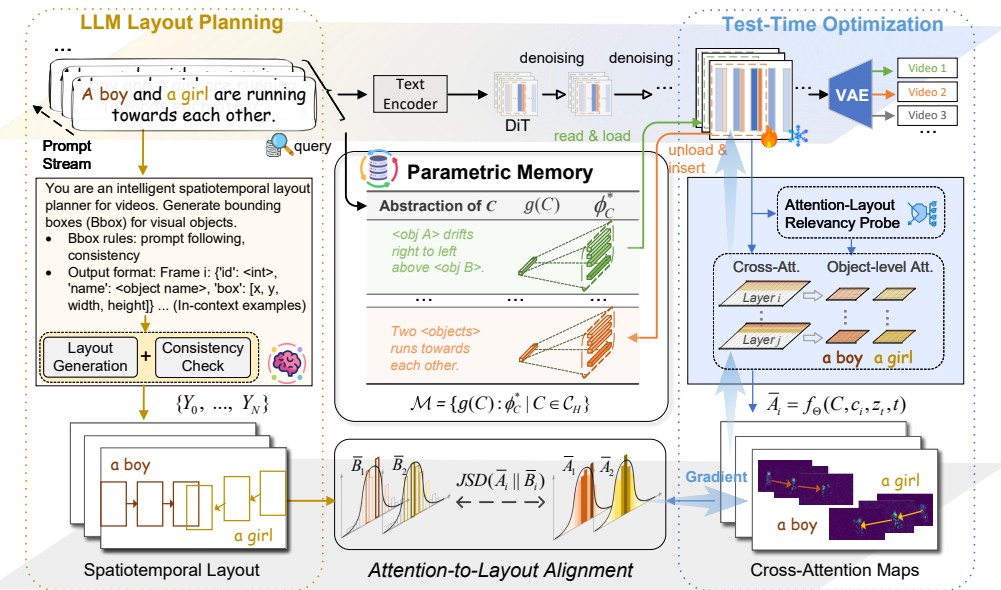

Figure 2: Overview of the **TTOM** framework for compositional text-to-video generation. A stream of text prompts is first fed into LLMs for spatial-temporal layout planning. Meanwhile, a denoising sampling process of video foundation models is performed, in which cross-attention maps are extracted, followed by test-time optimization for alignment. Historical optimization context is maintained by the parametric memory.

et al., 2024; Lin et al., 2023; Feng et al., 2025) through attention control (Chen et al., 2024; Xie et al., 2023; He et al., 2023; Wang et al., 2024; He et al., 2025) or latent modification (Qi et al., 2024; Wu et al., 2024; Lian et al., 2023a; Yang et al., 2024a; Wang et al., 2025c; Lian et al., 2023b), typically in a training-free manner. However, direct intervention on attention maps or latents may compromise visual quality. Moreover, the per-sample paradigm treats test samples in isolation, disregarding informative historical context.

**Test-Time Optimization**. TTO adapts a trained model to test instance(s) by performing optimization during inference, which has been widely applied in domain adaptation (Li et al., 2016; Sun et al., 2020; Shu et al., 2022) and alignment (Kim et al., 2025; Li et al., 2025). Existing approaches can be categorized into three groups: 1) optimizing intermediate representations (Kim et al., 2025; Li et al., 2025), 2) prompt tuning (Shu et al., 2022), and 3) optimizing model parameters (Schneider et al., 2020; Liang et al., 2020; Shu et al., 2021; Zhang et al., 2022; Gandelsman et al., 2022). In visual generation, recent work (Lian et al., 2023b; Wu et al., 2024) optimizes latents guided by layouts, yielding significant gains. However, these methods perform optimization on a per-sample basis and subsequently discard the results, thereby neglecting historical context.

**Memory Mechanism**. To track and model historical contexts, existing Test-Time Training (TTT) (Sun et al., 2024) work designs memory mechanisms instantiated with inserted layers, *i.e.*, TTT layers. The memory can be updated linearly (Sun et al., 2020) or in a local temporal window (Wang et al., 2025b), by optimizing TTT layers via a self-supervised learning loss (He et al., 2022). Despite the remarkable advance, they often consider intra-sample context (*e.g.*, frames in video) (Wang et al., 2025b; Dalal et al., 2025) and assume the independence among samples. Besides, the memory has a fixed size and does not support flexible operations.

## 3 METHODOLOGY

In this section, we introduce the proposed TTOM framework for compositional T2V, as shown in Fig. 2. Given a text prompt describing a compositional scene (*e.g.*, including multiple objects, attributes, numeracy, and relationships), we first conduct spatial-temporal layout (STL) planning

driven by LLMs, and obtain the bbox sequence for each object (Sec. 3.1). Subsequently, we perform test-time optimization to achieve attention-to-layout alignment, making the generated video follow the explicit layout guidance (Sec. 3.2). To take advantage of history context, we record previous optimization results with a parametric memory structure, and keep updating and reusing it as the sequential inference goes, *i.e.*, a ceaseless stream of user prompts is fed into the model (Sec. 3.3).

### 3.1 LLM-DRIVEN SPATIAL-TEMPORAL LAYOUT PLANNING

**Layout Representation**. We represent a video-level STL with a collection of object-level layouts $\{Y_0, ..., Y_N\}$ where $N$ denotes the number of objects. Each layout $Y_i$ is a tuple $(c_i, B_i)$, consisting of an object phrase $c_i$ and its associated bounding box sequence $B_i = [\mathbf{b}_{s_i}, \ldots, \mathbf{b}_{e_i}]$. The object phrase $c_i$ is extracted from the user prompt $C$ (*e.g.*, the underlined text in "*A vibrant red balloon drifts right to left above a grand statue*."). Each $\mathbf{b}$ is a 4-dimensional coordinate vector, while $s_i$ and $e_i$ denote the start and end frame indices of the object's temporal occurrence, respectively.

**Text-to-Layout Generation**. Prior work has shown that LLMs possess strong spatial-temporal dynamics understanding abilities (Qu et al., 2021; 2023; Feng et al., 2023; Qu et al., 2024a), and capture physical properties (Lian et al., 2023b), such as gravity, elasticity, and perspective projection, which align closely with real-world physics. Inspired by it, we prompt LLMs to generate SPLs given user prompts via in-context learning (Dong et al., 2022; Qu et al., 2025a). To enhance scene comprehension, the LLM is first instructed to produce descriptions of object motions and camera behaviors, followed by the corresponding layout generation. Finally, a verification step ensures spatial and temporal consistency of object phrases and bounding box sequences, with corrections applied when discrepancies are detected.

### 3.2 TEST-TIME OPTIMIZATION FOR LAYOUT-TO-VIDEO GENERATION

In this section, we first discuss the relation between intermediate attention maps of diffusion models and generated videos, and conduct a probe experiment to verify the relevance. On the basis of the relevance, we propose a model-agnostic controllable generation strategy by optimizing attention-to-layout alignment during inference, *i.e.*, **Test-Time Optimization (TTO)**.

**Attention-Layout Relevance Probe in Diffusion Transformers**. Prior work (Hertz et al., 2022) has demonstrated the relevance between cross-attention maps and spatial layouts, which is exploited for video editing (Qi et al., 2023) and controllable generation (Xie et al., 2023). However, with the architecture shift from UNets (Ronneberger et al., 2015) to Diffusion Transformers (DiTs) (Peebles & Xie, 2023) due to its stronger scalability, the recent study (Wang et al., 2025c) pointed out their difference, *i.e.*, the cross-attention maps in DiTs distribute more evenly than those in UNets, which may cause the ineffectiveness of previous methods due to the weak attention-layout relevance.

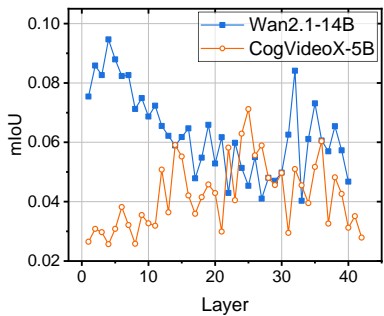

Figure 3: Attention-layout overlap (evaluated by mIoU (Everingham et al., 2010) over 200 prompts) between cross-modal attention maps extracted from each layer of foundation models and segmentation maps detected from generated videos by GroundingDINO (Liu et al., 2024) + SAM 2 (Ravi et al., 2024).

Motivated by it, we propose a probe strategy to assess attention-layout relevance across different layers: 1) preparing a set of prompts describing a common scene in which an object performs a motion. 2) generating videos given the prompts, and meanwhile, extracting text-video cross-attention maps. Specifically, we first extract the cross-attention map[1] $A_i \in \mathbb{R}^{\tau \times h \times w \times |c_i|}$ from each layer for the object phrase $c_i$ from a text prompt $C$, where $|c_i|$ denotes the token number of $c_i$. $\tau$, $h$, and $w$ refer to the frame number, height, and width of the video latent, respectively. Afterwards, we perform average pooling along the text dimension and obtain $\bar{A}_i \in \mathbb{R}^{\tau \times h \times w}$. This process is formulated as:

$$\bar{A}_i = f_\Theta(C, c_i, z_t, t), \tag{1}$$

---

[1]For foundation models (Yang et al., 2024b) without cross-attention layers, we extract those parts in self-attention maps where text-vision interaction happens.

where $z_t$ and $t$ represent the latent and timestep, respectively. $\Theta$ denotes the parameters of all layers before the attention map in a DiT. 3) applying GroundingDINO (Liu et al., 2024) to detect the related bboxes and SAM 2 (Ravi et al., 2024) for segmentation. And 4) computing the overlap between segmentation maps and attention maps $\bar{A}_i$ in different layers to quantify relevance. The result shown in Fig. 3 indicates a remarkable variance of attention-layout relevance among different layers.

**Attention-to-Layout Alignment for Controllable Generation**. As discussed above, attention maps in certain layers of DiTs reflect a strong relevance to the layout of the final generated videos. Motivated by this observation, we propose a test-time optimization approach to align the high-relevance attention maps with the layout generated by LLMs for controllable generation. Specifically, we first smooth each spatial layout $\mathbf{b}_i$ in $B_i$ with a Gaussian kernel and transform it into a soft mask $\bar{\mathbf{b}}_i$, and then calculate the Jensen–Shannon divergence (JSD) between attention maps and soft masks of $N$ objects as the loss function:

$$L_{align} = \frac{1}{N} \sum_{i=1}^{N} JSD(\bar{A}_i \| \bar{B}_i), \tag{2}$$

We insert new parameters $\phi$ into VFMs and get $\Theta = \theta \cup \phi$, calculate the gradient of the loss function *w.r.t.* the newly introduced parameters, *i.e.*, $\frac{\partial L_{align}}{\partial \phi}$, and update them with an optimizer and get $\phi^*$ finally. Compared with prior work (Lian et al., 2023b) that optimizes the latent $z_t$ at denoising timestep $t$ by $\frac{\partial L_{align}}{\partial z_t}$, we have the following strengths: 1) optimizing $\phi$ for alignment avoids possible distribution collapse caused by direct intervention to the latents. And 2) the updated $\phi^*$ memorizes layout patterns for specific compositional scenes, holding a potential for future re-usage. For example, if a similar text prompt comes in the future, a simple and efficient strategy is to directly load $\phi^*$ to the VFM, and perform generation without any optimization, which motivates us to propose a memory mechanism as follows.

## 3.3 PARAMETRIC MEMORY

To maintain historical contexts of test-time optimization and support future reuse, we present the parametric memory mechanism.

**Memory Structure**. We define a memory structure as a collection of key-value pairs:

$$\mathcal{M} = \{g(C) : \phi_C^* | C \in \mathcal{C}_H\}, \tag{3}$$

where $\mathcal{C}_H$ is a set of history prompts. The function $g(\cdot)$ consists of scene abstraction, text embedding, and indexing precedes. Specifically, for a prompt "*A vibrant red balloon drifts right to left above a grand statue.*", we first abstract it into "*<object A> drifts right to left above <object B>.*", and then extract its text feature (Radford et al., 2021) and index it.

**Memory Operations**. Based on the memory structure, we define the following basic operations to perform interaction between the video foundation model and the memory.

• *Insert*. For a new text prompt $C_j$, we first retrieve from the memory. If no matched item, we perform test-time optimization and insert a new item $(g(C_j), \phi_{C_j}^*)$ to $\mathcal{M}$.

• *Read*. If the matched item(s) exist, we read and load the corresponding parameters into the foundation model. Next, we have two options: 1) directly generate a video without optimization, and 2) continue optimization with the loaded parameters as initialization.

• *Update*. After the loaded parameters are further optimized, we unload them and update the corresponding matched items in the memory.

• *Delete*. If the total number of items in $\mathcal{M}$ exceeds its maximum capacity, we delete those items that are least frequently used.

**Discussion**. Compared with prior methods (Lian et al., 2023b; He et al., 2025) treating each sample independently, the advantages of the proposed memory mechanism lie in: 1) *Superiority*. The history optimization maintained in the memory provides abundant scene knowledge, serving as good initialization for TTO. 2) *Personalization*. In practice, we can maintain a user-specific memory for each user, track their historical requests, and model the intention for better personalized

Table 1: Evaluation results of compositional text-to-video generation on T2V-CompBench (Sun et al., 2025), reported over 7 categories and the overall average (Avg.).

| Model | Avg. | Motion | Num | Spatial | Con-attr | Dyn-attr | Action | Interact |
|---|---|---|---|---|---|---|---|---|
| *Commercial* | | | | | | | | |
| Pika-1.0 | 0.3752 | 0.2234 | 0.3870 | 0.4650 | 0.5536 | 0.0128 | 0.4250 | 0.5198 |
| Gen-3 | 0.4094 | 0.2754 | 0.2306 | 0.5194 | 0.5980 | 0.0687 | 0.5233 | 0.5906 |
| Dreamina 1.2 | 0.4689 | 0.2361 | 0.4380 | 0.5773 | 0.6913 | 0.0051 | 0.5924 | 0.6824 |
| Kling-1.0 | 0.4630 | 0.2562 | 0.4413 | 0.5690 | 0.6931 | 0.0098 | 0.5787 | 0.7128 |
| *Diffusion Unet based* | | | | | | | | |
| ModelScope | 0.3468 | 0.2408 | 0.1986 | 0.4118 | 0.5148 | 0.0161 | 0.3639 | 0.4613 |
| + LVD | 0.3912 | 0.2457 | 0.2008 | 0.5405 | 0.5439 | 0.0171 | 0.3802 | 0.4502 |
| Show-1 | 0.3676 | 0.2291 | 0.3086 | 0.4544 | 0.5670 | 0.0115 | 0.3881 | 0.6244 |
| VideoTetris | 0.4097 | 0.2249 | 0.3467 | 0.4832 | 0.6211 | 0.0104 | 0.4839 | 0.6578 |
| T2V-Turbo-V2 | 0.4317 | 0.2556 | 0.3261 | 0.5025 | 0.6723 | 0.0127 | 0.6087 | 0.6439 |
| *DiT based* | | | | | | | | |
| Open-Sora 1.2 | 0.3851 | 0.2468 | 0.3719 | 0.5063 | 0.5639 | 0.0189 | 0.4839 | 0.5039 |
| Open-Sora-Plan v1.3.0 | 0.3670 | 0.2377 | 0.2952 | 0.5162 | 0.6076 | 0.0119 | 0.4524 | 0.4483 |
| CogVideoX-5B | 0.4189 | 0.2658 | 0.3706 | 0.5172 | 0.6164 | 0.0219 | 0.5333 | 0.6069 |
| + DyST-XL | 0.5081 | 0.2712 | 0.3969 | 0.6110 | 0.8696 | 0.0221 | 0.7321 | 0.6536 |
| + LVD | 0.4739 | 0.3291 | 0.3825 | 0.5274 | 0.7534 | 0.0219 | 0.6826 | 0.6204 |
| + Ours | 0.5632 | 0.4351 | 0.5081 | 0.6173 | 0.8782 | 0.0341 | 0.7191 | 0.7502 |
| %Improve. | +34.45 | +63.69 | +37.10 | +19.35 | +42.47 | +55.71 | +34.84 | +23.61 |
| Wan2.1-14B | 0.5314 | 0.2696 | 0.5113 | 0.5709 | 0.8369 | 0.0570 | 0.7504 | 0.7239 |
| + LVD | 0.5439 | 0.2864 | 0.4707 | 0.5753 | 0.8610 | 0.0829 | 0.8107 | 0.7201 |
| + Ours | 0.6155 | 0.4922 | 0.5881 | 0.6275 | 0.8982 | 0.1182 | 0.8152 | 0.7691 |
| %Improve. | +15.83 | +82.57 | +15.02 | +9.91 | +7.32 | +107.37 | +8.64 | +6.24 |

generation (Wang et al., 2025a; Zhao et al., 2025; 2026). 3) **Efficiency**. For those prompts similar to historical ones, we can skip optimization and directly read and load items into the foundation model for inference. 4) **Scalability**. We can scale the memory up in two dimensions: increasing the item-level capacity (*i.e.*, enlarging the history context window) could memorize more historical optimization information; and increasing the number of per-item parameters $\phi^*$ may capture more scene patterns in each TTO.

# 4 EXPERIMENTS

## 4.1 EXPERIMENTAL SETUP

**Foundation Models**. We integrate our method into two representative T2V models: CogVideoX-5B (Yang et al., 2024b) and Wan2.1-14B (Wan et al., 2025). For CogVideoX-5B, each clip contains 49 frames at 8 FPS, whereas for Wan2.1-14B, each clip contains 81 frames at 16 FPS. In the Spatial–Temporal Layout Planning stage, we leverage the OpenAI GPT-4o model.

**Streaming Video Generation Setting**. To align with practical video creation scenarios where user prompts arrive sequentially, we propose modeling historical streaming contexts. For a fair comparison with prior methods that treat test samples independently, we introduce a *test-time independence* setting. Specifically, we generate 200 prompts covering common compositional scenes as *pseudo-training data*, via GPT-4o, and subsequently generate videos from these prompts while applying TTOM to construct a memory. During inference, entries can be retrieved and loaded from the memory; however, inserting or updating the memory with optimization results from test samples is prohibited.

**Implementation Details**. We employ the lightweight LoRA (Low-Rank Adaptation) (Hu et al., 2022) method to introduce new parameters to enable test-time optimization for controllable generation. The optimized LoRA parameters are inserted into or used to update the parametric memory. Specifically, LoRA weights are injected into the Query, Key, Value, and Output projection layers of each cross-attention block in DiT architectures, with the LoRA rank set to 32. This optimization is applied only to the first five denoising steps of the diffusion process, using the AdamW optimizer with a learning rate of 1e-4. Unless otherwise noted, all other hyperparameters follow the default settings of the respective backbone models. For fair comparison, we also re-implement the LVD method (Lian et al., 2023b) on the same DiT-based architecture. The layouts and selected layers used for LVD exactly

Table 2: Evaluation results on the semantic categories of VBench.

| Model | Obj. Class | Multi-Obj. | Human Act. | Color | Spatial Rel. | Scene | Appear. Style | Temp. Style | Overall Cons. |
|---|---|---|---|---|---|---|---|---|---|
| CogVideoX-5B | 0.8342 | 0.6728 | 0.9760 | 0.8840 | 0.7943 | 0.5328 | 0.2468 | 0.2542 | **0.2742** |
| + LVD | 0.8661 | 0.6782 | 0.9740 | 0.8802 | 0.8014 | 0.5406 | 0.2472 | 0.2509 | 0.2609 |
| + Ours | **0.9486** | **0.7952** | **0.9820** | **0.9246** | **0.8215** | **0.5682** | **0.2550** | **0.2576** | 0.2678 |
| Wan2.1-14B | 0.8628 | 0.6958 | 0.9540 | 0.8859 | 0.7539 | 0.4575 | 0.2264 | 0.2319 | **0.2591** |
| + LVD | 0.9083 | 0.7286 | 0.9620 | 0.8832 | 0.7547 | 0.5135 | 0.2280 | 0.2407 | 0.2507 |
| + Ours | **0.9921** | **0.8216** | **0.9800** | **0.9289** | **0.8074** | **0.5286** | **0.2305** | **0.2444** | 0.2542 |

match those used for our TTO version, while all other training hyperparameters follow the original LVD paper.

**Benchmarks and Evaluation Metrics**. We evaluate our approach on two large-scale public benchmarks, T2V-CompBench (Sun et al., 2025) and VBench (Huang et al., 2024). T2V-CompBench targets compositional T2V with about 1,400 prompts across seven categories, including consistent and dynamic attribute binding, spatial and motion relations, action binding, object interactions, and generative numeracy. Its evaluation combines MLLM-based scoring, detection-based measures, and tracking-based indicators to provide fine-grained assessment of compositional abilities. VBench offers a broad evaluation suite, decomposing video generation quality into 16 dimensions, including subject and background consistency, temporal flickering, motion smoothness, dynamic degree, aesthetic and imaging quality, as well as object class, multiple objects, human action, color, spatial relationship, scene, appearance style, temporal style, and overall consistency. Each dimension contains roughly 100 text prompts with automatic evaluation scripts and human preference annotations. We follow the official toolkits of both benchmarks and report corresponding scores.

**Baselines**. We conduct a comprehensive comparison of our approach on T2V-CompBench against a diverse set of baselines across three categories: commercial solutions (Pika-1.0 (Pika, 2024), Gen-3 (Runway, 2024), Dreamina-1.2 (CapCut, 2024), Kling-1.0 (Kuaishou, 2024)); diffusion U-Net–based methods (ModelScope (Wang et al., 2023), Show-1 (Zhang et al., 2024), VideoTetris (Tian et al., 2024), T2V-Turbo-V2 (Li et al., 2024), LVD (Lian et al., 2023a)); and DiT-based models (Open-Sora-1.2 (HPC-AI Tech, 2024), Open-Sora-Plan-v1.3.0 (Lin et al., 2024), CogVideoX-5B (Yang et al., 2024b) with DyST-XL (He et al., 2025) and LVD, Wan2.1-14B (Wan et al., 2025) with LVD). Except for CogVideoX-5B with LVD, Wan2.1-14B, and Wan2.1-14B with LVD, for which we reproduce the results in our experiments, the scores of all remaining baselines are taken directly from the official T2V-CompBench paper (Sun et al., 2025).

## 4.2 PERFORMANCE COMPARISON

**Evaluation on Compositionality**. Table 1 presents the compositional evaluation results of our method compared to a wide range of baselines on T2V-CompBench. Across all metrics, our approach consistently surpasses the baseline models, demonstrating effectiveness and superiority. Notably, our method improves CogVideoX-5B by **34.45%** and Wan2.1-14B by **15.83%** in terms of overall average performance. The most pronounced gains are observed in Motion and Numeracy, with relative improvements of up to **63.69%** and **37.10%** on CogVideoX-5B, and **82.57%** and **15.02%** on Wan2.1-14B, respectively. These categories are known to be particularly challenging due to the need for precise modeling of dynamic object trajectories, object counts, and spatial-temporal coordination.

**Evaluation on Semantic Consistency**. Beyond compositionality, we further evaluate semantic consistency on VBench (Table 2). Here, our method also delivers clear improvements across multiple dimensions, including object classification accuracy, multi-object handling, and color/spatial relation fidelity. These aspects are crucial for ensuring that generated videos do not only satisfy individual attributes but also maintain coherent semantic integrity across frames. The gains in semantic consistency complement the compositionality improvements observed on T2V-CompBench, providing strong evidence that our method reinforces both the local correctness of attributes (*e.g.*, colors, actions, object categories) and the global contextual structure (*e.g.*, multi-object interactions and spatial layouts).

## 4.3 In-depth Analysis

To explore the efficacy of TTOM, we conduct extensive ablation studies and hyperparameter analyses. We first investigate the TTO and Memorization, followed by an in-depth study of key components, including continual TTO, loss function for attention-to-layout alignment, guidance timesteps, and optimization iteration.

**Test-Time Optimization and Memorization**. We conduct comparative experiments to evaluate the effectiveness of TTO and the memory mechanism, as reported in Tab. 3. The results show that TTO substantially improves motion quality in generated videos by **60.27%**, guided by LLM-planned spatiotemporal layouts. Moreover, incorporating historical context from memory yields an additional **13.91%** improvement.

Table 3: Ablation study for TTO and Memory on the motion category of T2V-CompBench (Sun et al., 2025).

| TTO | Memory | **Motion** |
|-----|--------|--------|
| ✗ | ✗ | 0.2696 |
| ✓ | ✗ | 0.4321 |
| ✓ | ✓ | 0.4922 |

**Continual Test-Time Optimization**. The parametric memory enables the model to exploit historical context for future inference. In practice, however, memory capacity is limited and cannot fully encompass world knowledge (*e.g.*, object motion patterns). Consequently, imperfect matching may arise between the current prompt and the retrieved items, leading to suboptimal parameter initialization. Therefore, continual optimization can help balance historical context and the current sample. To examine its effect, we conduct experiments in Tab. 4.

The results indicate that: 1) The memory mechanism is effective, as evidenced by substantial gains even without per-sample optimization. 2) With memory-based initialization, continual TTO further improves motion performance at the cost of additional test-time computation. And 3) increasing the number of memory entries as initialization provides richer context but may also introduce noise or irrelevant information, suggesting that more sophisticated fusion strategies are required for better leveraging context in the future. In summary, the proposed TTOM offers a flexible trade-off between video quality and efficiency, making it well-suited for complex practical scenarios.

Table 4: Ablation study for continual test-time optimization on the motion category of T2V-CompBench (Sun et al., 2025). Motion score and latency are used to evaluate motion quality and efficiency, respectively. Init.: initialization from memory with the average fusion of Top-$k$ matched entries. TTO: whether to perform test-time optimization.

| Init. | TTO | Top-$k$ | Motion↑ | Latency (s)↓ |
|-------|-----|---------|---------|--------------|
| ✗ | ✗ | – | 0.2696 | 425 |
| ✓ | ✗ | 5 | 0.4754 | 427 |
| ✓ | ✓ | 5 | 0.4846 | 627 |
| ✓ | ✗ | 10 | 0.4437 | 427 |
| ✓ | ✓ | 10 | 0.4705 | 627 |

**Loss Function for Attention-to-Layout Alignment**. To drive controllable generation guided by spatiotemporal layout, we propose a JSD loss to perform attention-to-layout alignment optimization during inference. To validate its effectiveness, we conduct experiments to compare it with two other variants: BCE loss (Sella et al., 2025) and Center-of-Mass (CoM) loss (Lian et al., 2023b). Results in Tab. 5 show that the proposed JSD loss achieves the best performance on motion and numeracy, demonstrating its effectiveness and universality.

**Pseudo-training Data Scale**. To simulate the streaming generation scenario and make a strictly fair comparison with prior methods, we first generate user prompts via GPT-4o to initialize the memory. Results in Tab. 6 show the influence of different numbers of pseudo-training data on the performance. We can see that more data induces better performance, due to the more abundant compositional patterns captured and saved into the memory as context.

Table 5: Comparison among different loss functions for attention-to-layout alignment, evaluated on the motion and numeracy categories of T2V-CompBench.

| Loss Function | Motion | Numeracy |
|---------------|--------|----------|
| CE Loss | 0.2912 | 0.5218 |
| CoM Loss | 0.3626 | 0.4697 |
| JSD Loss | 0.4321 | 0.5881 |

**Guidance Timesteps and Optimization Iteration**. Existing work (Choi et al., 2022; Pan et al., 2024) has shown that early denoising steps in diffusion models primarily define structure, while later steps progressively refine details. Motivated by this observation, we apply TTO only in the initial denoising steps and compare performance across different numbers

Table 6: Impact of pseudo-training data scale, evaluated on the motion category of T2V-CompBench.

Table 7: Impact of number of timesteps for optimization during the early stage of denoising sampling.

Table 8: Impact of the number of test-time optimization iterations per denoising sampling timestep.

| #Pseudo-training Data | Motion |
|---|---|
| 50 | 0.3519 |
| 100 | 0.4344 |
| 150 | 0.4751 |
| 200 | 0.4922 |

| #Timestep | Motion |
|---|---|
| 1 | 0.2730 |
| 3 | 0.3692 |
| 5 | 0.4321 |
| 7 | 0.4296 |

| #Iteration | Motion |
|---|---|
| 4 | 0.3789 |
| 8 | 0.4321 |
| 12 | 0.4130 |
| 16 | 0.3547 |

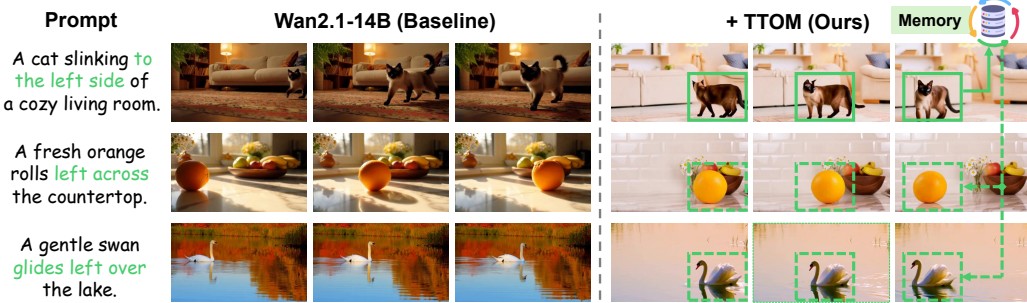

Figure 4: Qualitative results of motion pattern transfer with memory. Solid arrows indicate insert or update operations, while dotted arrows represent reading and loading parameters from memory into foundation models for inference.

of steps, as presented in Tab. 7. Within each step, we further examine the impact of varying the number of optimization iterations, reported in Tab. 8. The results reveal clear saturation points for both timesteps and optimization iterations, beyond which performance degrades. This degradation may arise because attention maps encode not only layout but also entangled information, and excessive alignment optimization may disrupt other essential signals.

**Qualitative Results**. Fig. 1 shows the qualitative comparison between Wan2.1-14B and our method based on which. To further delve into the impact of the proposed memory mechanism, we show the qualitative results of motion pattern transfer from memory to foundation models in Fig. 4. More qualitative results can be found in the Appendix.

## 5 CONCLUSION

In this work, we tackle the compositional limitations of Video Foundation Models and introduce TTOM, a training-free framework for aligning video generation with spatiotemporal layouts at inference. By integrating layout-guided optimization with a parametric memory mechanism, TTOM enables consistent streaming generation and supports flexible operations on memory, such as insertion, update, and retrieval of historical contexts. Our experimental analysis shows that TTOM not only improves alignment in compositional scenarios, but also disentangles compositional world knowledge, leading to strong transferability and generalization. Extensive experiments on T2V-CompBench and VBench confirm its effectiveness, scalability, and practicality, establishing TTOM as a versatile solution for enhancing compositional video generation on the fly.

## ETHICS STATEMENT

Our work on compositional video generation holds significant promise for democratizing creative tools, enhancing accessibility, streamlining media production, and advancing intuitive human–AI collaboration. At the same time, it raises critical concerns, including potential misuse for misinformation or manipulation, biases in generated content, job displacement in creative industries, and the environmental costs of intensive computation. Addressing these challenges requires careful dataset curation, robust privacy safeguards, systematic bias mitigation, responsible deployment strategies, and sustained engagement with diverse stakeholders.

## REPRODUCIBILITY STATEMENT

We have taken several steps to ensure the reproducibility of our work. A link (`https://ttom-t2v.github.io/`) to the source code is provided, enabling replication of our implementation. The main text and appendix together provide comprehensive descriptions of the model design, optimization procedure, and evaluation protocol. These resources collectively ensure that readers can reproduce and validate our experimental results.

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

## A THE USE OF LARGE LANGUAGE MODELS

We declare that we used large language models (LLMs) to assist in refining this manuscript, specifically for grammar checking, language polishing, and enhancing textual clarity and fluency. We also employed LLMs in a limited capacity for minor debugging and syntactic corrections of code snippets.

## B ADDITIONAL EXPERIMENTAL RESULTS

### B.1 OVERALL AND QUALITY EVALUATION ON VBENCH

Tables 9 and 10 report the overall and quality dimension results on VBench. Our method achieves consistently higher scores than both the original backbones and the LVD-enhanced variants, confirming its effectiveness in improving perceptual quality. In the quality dimensions (Table 10), our method matches or surpasses the strongest baseline on five dimensions, generating visually sharper and temporally smoother videos. These findings underline that our method yields a better balance between perceptual fidelity and text adherence than all compared baselines.

Table 9: Overall evaluation results on VBench.

| Model | Total | Quality | Semantic |
|---|---|---|---|
| CogVideoX-5B | 0.8201 | 0.8272 | 0.7917 |
| + LVD | 0.7992 | 0.8010 | 0.7921 |
| + Ours | **0.8318** | **0.8314** | **0.8332** |
| Wan2.1-14B | 0.8369 | 0.8559 | 0.7611 |
| + LVD | 0.8106 | 0.8186 | 0.7788 |
| + Ours | **0.8492** | **0.8573** | **0.8166** |

Table 10: Quality dimension results on VBench.

| Model | Sub. Cons. | B.g. Cons. | Temp. Flick. | Motion Smooth | Dyn. Deg. | Aesth. Qual. | Imag. Qual. |
|---|---|---|---|---|---|---|---|
| CogVideoX-5B | 0.9656 | 0.9681 | 0.9853 | 0.9815 | **0.5616** | 0.6207 | **0.6534** |
| + LVD | 0.9532 | 0.9602 | 0.9823 | 0.9830 | 0.4672 | 0.6008 | 0.5782 |
| + Ours | **0.9682** | **0.9773** | **0.9860** | **0.9855** | 0.5426 | **0.6273** | 0.6527 |
| Wan2.1-14B | **0.9752** | 0.9809 | 0.9946 | 0.9830 | 0.6546 | 0.6607 | **0.6943** |
| + LVD | 0.9531 | 0.9719 | 0.9942 | 0.9835 | 0.5476 | 0.6437 | 0.5602 |
| + Ours | 0.9577 | **0.9901** | **0.9952** | **0.9895** | **0.6602** | **0.6619** | 0.6837 |

### B.2 ADDITIONAL QUALITATIVE RESULTS

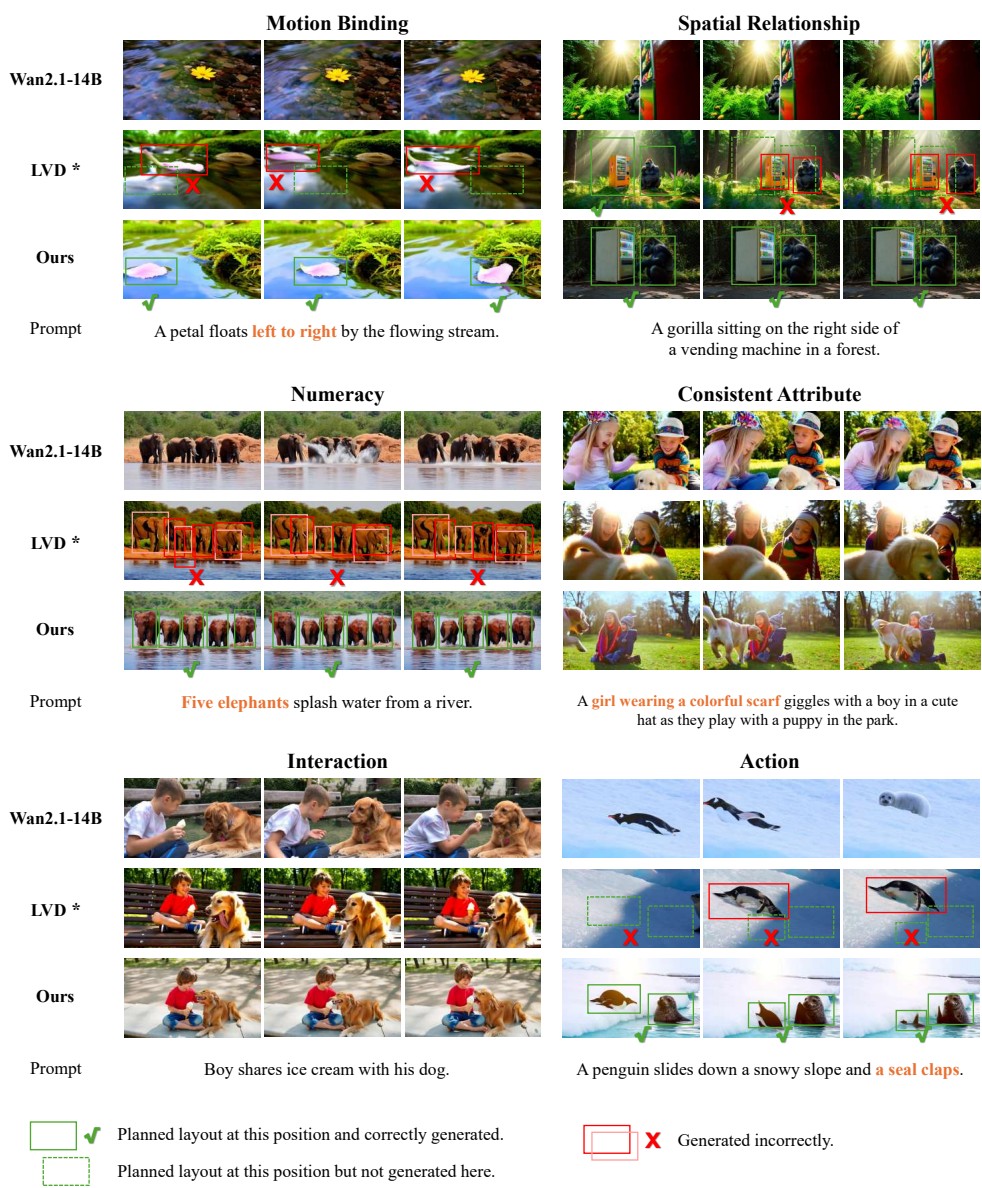

Figure 5: Qualitative comparison between the foundation, the baseline, and our method on T2V-CompBench.

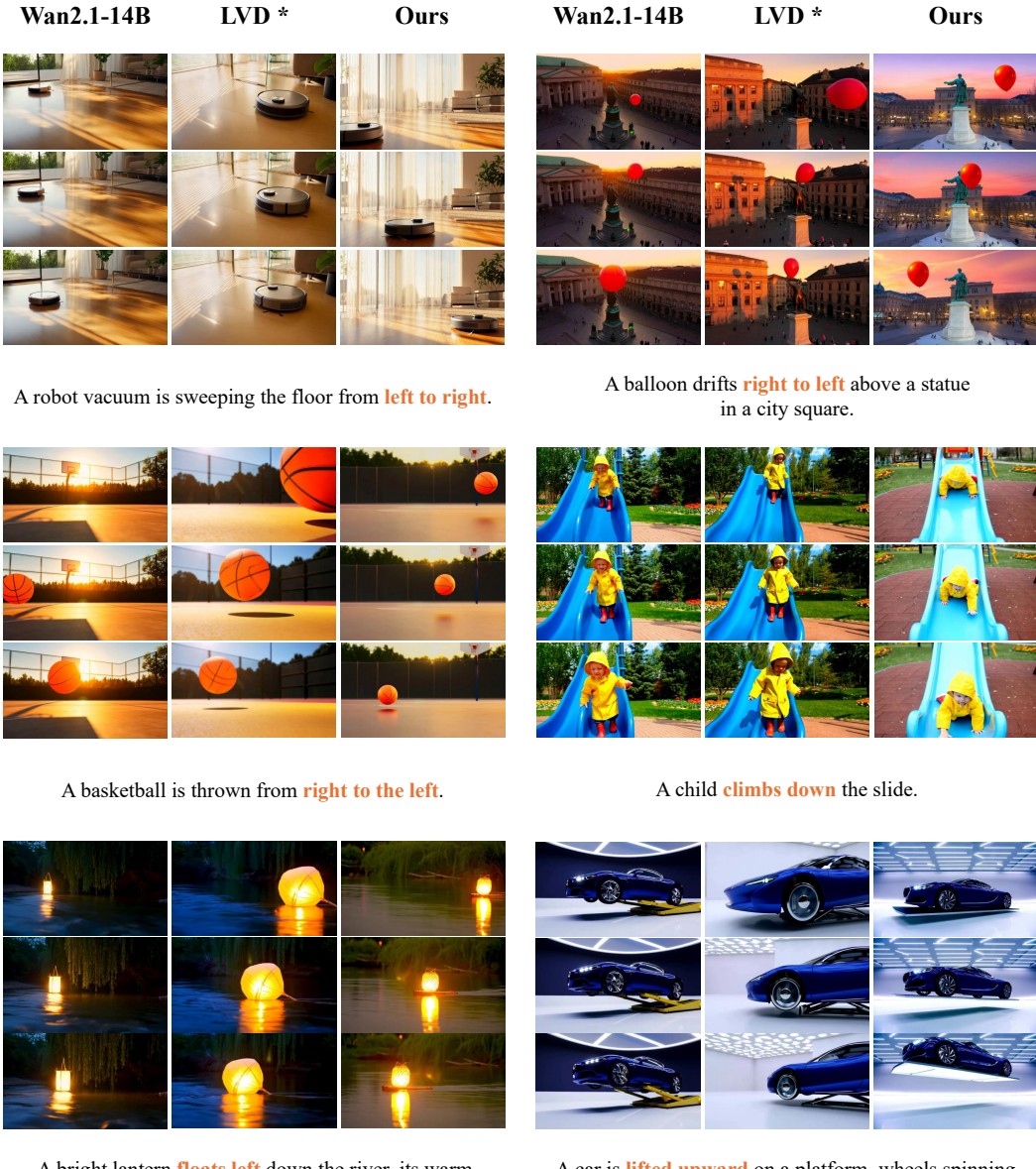

Figure 6: More Qualitative comparison between the foundation, the baseline, and our method on T2V-CompBench.

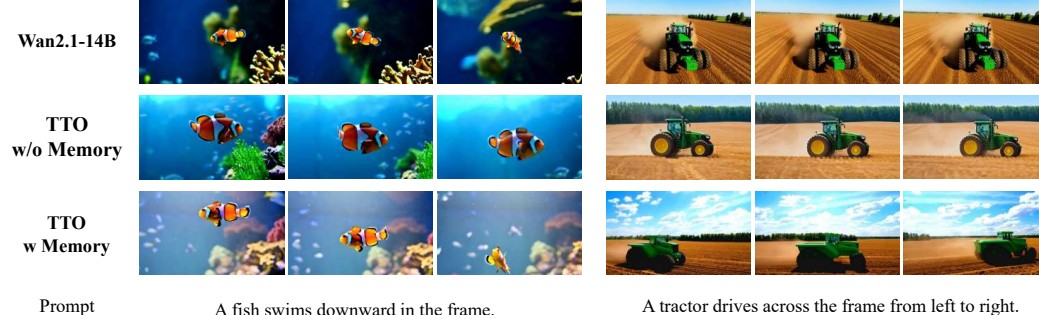

Prompt: A fish swims downward in the frame. — A tractor drives across the frame from left to right.

(a) These two examples show that adding memory initialization to TTO leads to better dynamic performance. Stand-alone TTO is constrained by the layout—if the LLM generates a layout with small motion amplitudes, the movement direction may not be clearly visible. However, with memory initialization, the motion dynamics in the frames are significantly enhanced.

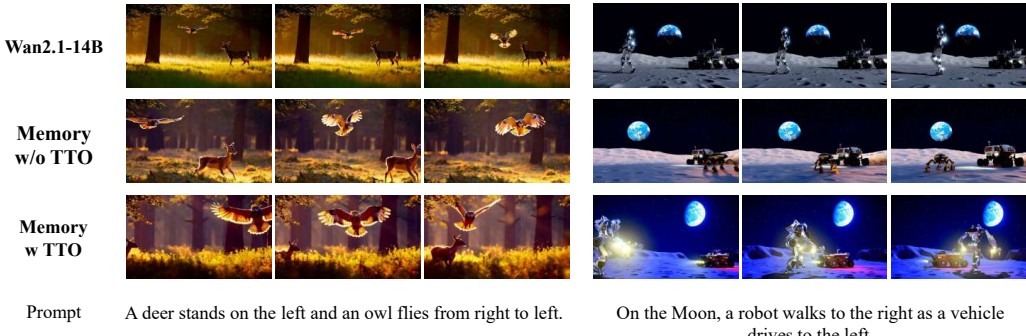

Prompt: A deer stands on the left and an owl flies from right to left. — On the Moon, a robot walks to the right as a vehicle drives to the left.

(b) These two examples show that adding TTO on top of memory initialization yields better constraint behavior. Some prompts are relatively rare in the memory, and in extreme cases may not match a perfectly aligned motion pattern—for example, the two subjects performing different actions in these examples. Applying TTO in such cases leads to much stronger consistency.

Figure 7: Qualitative comparison to the impact of memory and TTO.

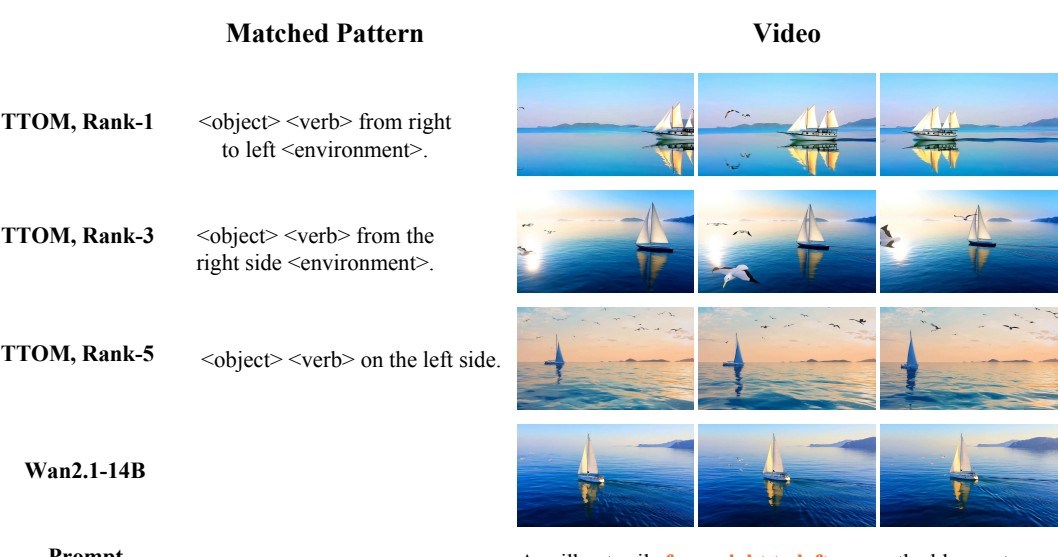

Figure 8: Qualitative comparison when loading items with different retrieval ranks in memory.

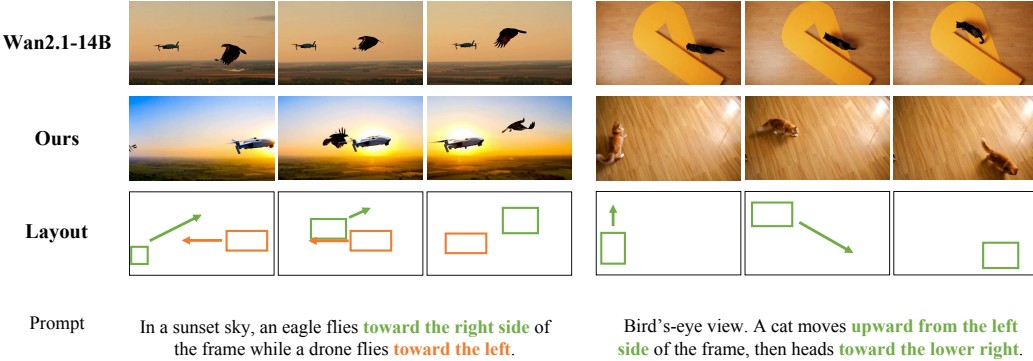

Prompt: In a sunset sky, an eagle flies **toward the right side** of the frame while a drone flies **toward the left**.

Bird's-eye view. A cat moves **upward from the left side** of the frame, then heads **toward the lower right**.

Figure 9: Qualitative results on complex cases involving multiple objects or non-linear motion.

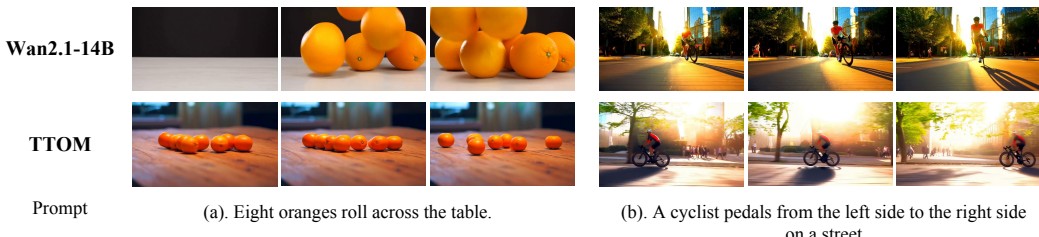

Prompt: (a). Eight oranges roll across the table.

(b). A cyclist pedals from the left side to the right side on a street.

Figure 10: Failure cases of TTOM.

## C  PROMPT DESIGN FOR LLM PLANNING

This appendix details the system prompts used in the first stage (LLM Planning) of our pipeline. The planning process involves prompt enrichment, metadata extraction, layout generation, and a dedicated verification stage to ensure consistency.

### C.1  PROMPT ENRICHMENT

The following system prompt is used to expand short user inputs into detailed video descriptions.

```
You are part of a team of bots that creates videos. You work with an
    assistant bot that will draw anything you say in square brackets.

For example, outputting "a beautiful morning in the woods with the sun
    peaking through the trees" will trigger your partner bot to output a
    video of a forest morning, as described. You will be prompted by
    people looking to create detailed, amazing videos. The way to
    accomplish this is to take their short prompts and make them
    extremely detailed and descriptive.

There are a few rules to follow:
- You will only ever output a single video description per user request.
- When modifications are requested, you should not simply make the
    description longer. You should refactor the entire description to
    integrate the suggestions.
- Other times the user will not want modifications, but instead want a
    new image. In this case, you should ignore your previous
    conversation with the user.
- Video descriptions must have the same num of words as examples below.
    Extra words will be ignored.
- Always preserve the original prompt's core content. For example, if
    the original prompt describes a motion such as "to the left side",
    please explicitly include both the starting location and the
    destination. For instance, revise the description to say "from the
    right side to the left side". Integrate all given details faithfully
    into your expanded description.
```

### C.2  METADATA EXTRACTION

This prompt extracts structured object information from the enriched description.

```
You are an intelligent assistant that extracts **object metadata** from
    descriptive paragraphs.

You will receive a descriptive paragraph. Your job is to:
1. Identify all distinct **objects** in the paragraph. Objects must be
    **concrete physical entities** that play a **visible or active
    role**.
2. Do **not** extract parts or attributes (e.g. "wings", "eyes").
3. Only extract objects that literally appear in the original_prompt.
4. For each object:
   - Assign a unique "object_id".
   - Provide the "object" canonical name.
   - Under "object_phrases", list quoted phrases (min 3 words).
   - Under "object_words", list single words corresponding to phrases.
5. If multiple objects with the same name appear, keep the name the
    same, but assign different ids.

Respond in the specified JSON format.

Guidelines:
- Exclude abstract ideas, emotions, locations, and background elements.
- Do not include synonyms unless literally mentioned.
```

```
- Strictly preserve original case and formatting.
```

## C.3 LAYOUT GENERATION

This prompt generates bounding box sequences based on the prompt and metadata.

```
You are an intelligent bounding box generator for videos.

You will receive:
1. original_prompt (core motion intent)
2. converted_prompt (visual details)
3. object metadata (entities list)

Your task: generate realistic bounding boxes (720x480) per frame (1-6)
    for main objects.

Bounding box rules:
- Follow motion path from original_prompt exactly.
- Format: {'id': <int>, 'name': <object name>, 'box': [x, y, width,
    height]}.
- **Numeracy**: If multiple objects of the same type exist, assign
    unique IDs and generate separate boxes.
- Match box size to camera framing (Close-ups: >=500x300).
- Avoid overlap unless implied.

Output format:
- Reasoning paragraph.
- Frames 1 to 6: list of bounding box dicts.
- Background keyword.
```

## C.4 VERIFICATION

To ensure the robustness of the generated plan, we employ two distinct verification steps: one for textual metadata and one for spatiotemporal layout.

### C.4.1 METADATA VERIFICATION PROMPT

This step corrects format errors and ensures linguistic guidelines are followed.

```
Please check if the extracted object metadata strictly follows the
    guidelines.
If there are any issues (e.g., inclusion of background elements,
    abstract parts, possessive pronouns like "its", or mismatches
    between "object_phrases" and "object_words"), correct them.

Return ONLY the corrected "objects" field. If no problems, return
    unchanged.

Rules:
- **Do not include possessive pronouns** (e.g., "its", "his") in
    "object_phrases".
- **"object_words" must only contain base object words** (e.g., use
    "car" instead of "car's").
- Ensure "object_words" are strictly nouns representing whole entities,
    not attributes.
```

### C.4.2 LAYOUT VERIFICATION PROMPT

This step critiques the generated bounding boxes, focusing on **numerical accuracy** and **spatiotemporal consistency**.

```
You are a Quality Assurance expert for video layout generation.
You will receive:
1. The Object Metadata (containing the list of objects to be detected).
2. The Generated Layout (Frames 1-6).

Your task is to verify and correct the layout based on the following
    strict criteria:

1. **Numeracy Check (Critical)**:
   - Verify that the number of bounding boxes for each object category
    matches the count specified in the metadata.
   - Example: If the metadata lists "cat" with ID 0 and "cat" with ID 1
    (total 2 cats), every frame MUST contain exactly 2 boxes for "cat".
   - If a specific number is mentioned (e.g., "5 birds"), ensure there
    are 5 distinct IDs.

2. **Temporal Consistency**:
   - Motion must be smooth. Eliminate random jitter or sudden jumps in
    position unless the prompt describes erratic movement.
   - Ensure ID consistency: Object ID 0 in Frame 1 must correspond to
    the same entity in Frame 6.

3. **Size & Bounds**:
   - Objects should not randomly shrink/grow without perspective
    justification.
   - Ensure all boxes are within valid coordinates [0, 0, 720, 480].

If errors are found, regenerate the "frames" section with corrected
    coordinates and counts.
Return the output in the exact same JSON format as the input.
```

## D  THEORETICAL ANALYSIS OF THE MEMORY MECHANISM

In this section, we provide a formal theoretical analysis of the proposed memory mechanism. We frame the mechanism not merely as a caching strategy, but as a *Manifold-Aware Initialization* scheme that accelerates convergence by leveraging the local smoothness of the task-parameter mapping. We further analyze the stability of the update rule to demonstrate its robustness against catastrophic forgetting.

### D.1  PROBLEM FORMULATION: TTO AS TRAJECTORY OPTIMIZATION

Let $\mathcal{T}$ denote the manifold of compositional video generation tasks (user prompts), and let $\Phi \subseteq \mathbb{R}^d$ represent the parameter space of the trainable modules (e.g., LoRA weights). We define a mapping $F : \mathcal{T} \to \Phi$ such that for any task $C \in \mathcal{T}$, there exists an optimal parameter set $\phi_C^*$ that minimizes the alignment objective $\mathcal{L}_{\text{align}}$:

$$\phi_C^* = \arg\min_{\phi \in \Phi} \mathcal{L}_{\text{align}}(\phi; C) \tag{4}$$

The standard Test-Time Optimization (TTO) process seeks to find $\phi_C^*$ via Gradient Descent starting from a generic initialization $\phi_0$. The memory $\mathcal{M}$ is defined as a discrete set of tuples $\{(k_i, v_i)\}_{i=1}^N$, where $k_i = g(C_i)$ is the semantic representation (key) of a historical task, and $v_i = \phi_{C_i}^*$ is the converged parameter state (value).

### D.2  CONVERGENCE ANALYSIS: THE INITIALIZATION ADVANTAGE

The efficiency of our method stems from the bounding of the optimization trajectory. We rely on the *Task-Parameter Smoothness Assumption*: we assume that if two tasks $C_i$ and $C_j$ are semantically close in the embedding space (i.e., $\|g(C_i) - g(C_j)\| < \epsilon$), their corresponding optimal parameters lie within a local neighborhood on the parameter manifold: $\|\phi_{C_i}^* - \phi_{C_j}^*\| < \delta$.

Under the assumption that the loss landscape $\mathcal{L}_{\text{align}}$ is locally $\mu$-strongly convex and $L$-smooth around the optimum (a standard assumption in fine-tuning analysis), the convergence rate of Gradient Descent at step $t$ is bounded by the initial distance to the solution:

$$\|\phi_t - \phi_C^*\|^2 \leq (1 - \eta\mu)^t \|\phi_{\text{init}} - \phi_C^*\|^2 \tag{5}$$

where $\eta$ is the learning rate.

In our proposed mechanism, the retrieval (Liu et al., 2018; Wen et al., 2024) function $R(\mathcal{M}, C)$ selects an initialization $\phi_{\text{mem}}$ such that:

$$\phi_{\text{init}} = \begin{cases} \phi_{\text{mem}} & \text{if } \exists (k, v) \in \mathcal{M} \text{ s.t. } \text{sim}(g(C), k) > \tau \\ \phi_{\text{random}} & \text{otherwise} \end{cases} \tag{6}$$

When a similar structural layout is retrieved, $\|\phi_{\text{mem}} - \phi_C^*\| \ll \|\phi_{\text{random}} - \phi_C^*\|$. Consequently, the number of steps $T$ required to reach an error tolerance $\epsilon$ is significantly reduced. This theoretical bound formally explains the reduction in inference latency and the enhanced motion scores observed in our empirical results.

### D.3 STABILITY AND ORTHOGONAL INFORMATION RETENTION

A critical challenge in online learning is *Catastrophic Forgetting*, where updates for a new task degrade performance on previous tasks. We prove that our mechanism is immune to this phenomenon via its discrete storage topology.

Unlike continual learning approaches that update a shared parameter set $\theta \leftarrow \theta - \nabla\mathcal{L}_{\text{new}}$, our memory update mechanism is additive and orthogonal. The update rule is formally defined as:

$$\mathcal{M}_{t+1} \leftarrow \mathcal{M}_t \cup \{(g(C_{\text{new}}), \phi_{\text{new}}^*)\} \tag{7}$$

Because each optimized parameter set $\phi^*$ is stored as a discrete entry effectively isolated in the memory bank, the gradient updates for task $C_{\text{new}}$ do not alter the values $v_i$ associated with historical keys $k_i$.

**Forgetting Dynamics.** The only information loss in the system is governed by the cache replacement policy (e.g., Least Frequently Used) which occurs only when the memory capacity $|\mathcal{M}|$ exceeds the maximum limit $K_{\text{max}}$. This ensures that the system prioritizes the retention of high-probability compositional structures, effectively approximating the optimal storage policy for the distribution of user prompts.

