# OpenReview forum: "TTOM: Test-Time Optimization and Memorization for Compositional Video Generation"
_ICLR.cc/2026/Conference — ICLR 2026 Poster_

### Official Review · Reviewer_HTdp · 2025-10-31

**Soundness:** 3
**Presentation:** 4
**Contribution:** 3
**Rating:** 8
**Confidence:** 3

**Summary:**

The paper proposes a method to enhance the alignment capability of compositional text-to-video generation during the inference phase through test-time optimization and a memory mechanism.

**Strengths:**

1. This paper introduces the TTOM framework, which combines test-time optimization with a parametric memory mechanism, enabling dynamic adjustment of model parameters at inference time and retaining historical optimization results for reuse with similar future prompts.
2. This approach not only improves text-video alignment in compositional scenarios but also supports continuous learning and personalized generation, demonstrating strong practicality and scalability. On mainstream benchmarks such as T2V-CompBench and VBench, the TTOM method significantly outperforms existing approaches across multiple key metrics (e.g., motion, numeracy, spatial relations), achieving overall improvements of up to 34% and 15% on large models like CogVideoX-5B and Wan2.1-14B, respectively, thereby validating its effectiveness and generalization capability.

**Weaknesses:**

1. The effectiveness of the method heavily relies on the quality of the spatiotemporal layouts generated by the LLM. If the bounding box sequences produced by the LLM are inaccurate, subsequent attention alignment optimization may fail or even introduce misleading guidance, compromising the quality of the generated content.

2. Test-time optimization requires extra gradient computations and parameter updates, which significantly increase inference latency, especially during continuous optimization and memory retrieval. Although the paper mentions that optimization is applied to the first five denoising steps of the diffusion process to partially mitigate this issue, its acceptability in real-world streaming generation scenarios still requires further validation.

**Questions:**

Please refer to the weaknesses.

---

> ### Author Response · Authors · 2025-11-21
> **Author Response to Reviewer HTdp**
>
> We sincerely thank the reviewer for the comprehensive evaluation and thoughtful feedback. Your constructive insights have significantly strengthened the clarity and quality of this manuscript. Our point-by-point responses are presented below.
>
> ---
>
> > (W1) the method heavily relies on the quality of the spatiotemporal layouts
>
> Thanks for raising this concern.
> Despite the weak layout planning ability of state-of-the-art video generative models [a, b], we found that **current LLMs could handle this task well**.
> We first conducted **human evaluation on layout planning** as shown in the table below. The results validate the reliability of LLM layout planning, and the proposed verification step could further improve the accuracy.
> | Verification | Success Rate for Layout Planning |
> |-------------|--------------|
> | -           | 87.1%        |
> | ✓           | **98.2%**        |
>
> We also adopted another lightweight LLM, Qwen3-Omni-Flash, for layout planning and made a comparison with GPT-4o. It shows our method could benefit from a stronger LLM (gpt-4o), but the lightweight LLM could also provide trustworthy layouts. It indicates that layout planning performed by current LLMs is reliable and we do not have to heavily depend on a very strong LLM.
>
> | LLM                 |  Verification | Video Generation Performance |
> |---------------------|--------------|-------------|
> | -                   |  -            | 31.64       |
> | gpt4o               |  ×            | 47.86       |
> | gpt4o               |  ✓            | **48.57**       |
> | Qwen3-Omni-Flash    |  ✓            | 47.29       |
>
> Furthermore, we also intentionally add noise to layouts and then perform generation with our model. The results verify the robustness of our method to noisy layouts.
>
> | Noise Type              | Performance |
> |-------------------------|-------------|
> | No noise                | 0.4857   |
> | Random perturbation     | 0.4821       |
>
> [a] Wan: Open and Advanced Large-Scale Video Generative Models. arXiv'25
>
> [b] CogVideoX: Text-to-Video Diffusion Models with An Expert Transformer. ICLR'25
>
> ---
>
> > (W2) inference latency brought by test-time optimization in real-world streaming generation scenarios
>
> Thanks for raising this concern about the efficiency and quality trade-off. As shown in Tab. 4, in fact, our method could achieve a promising trade-off, with **a remarkable performance gain of 76% (0.2696 to 0.4754), with only an additional 0.5% latency (425s to 427s)**, by comparing the first two rows. Even though we perform continuous TTO, the larger 47.5% latency is also acceptable compared with the performance improvement.
>
> In real-world scenarios, we could deploy a much larger memory to cover a broad range of creation scenarios collected by real users during the cold start phase, and then flexibly decide whether to perform continuous TTO according to the preference for efficiency or quality. Compared with future large throughput requests, the cold start phrase for memory deployment seems much more marginal.

---

> ### Author Response · Authors · 2025-11-25
> **Follow-up on Discussion**
>
> Dear Reviewer HTdp,
>
> Thank you again for your time and for the positive evaluation. We truly appreciate your careful review. We have addressed the concerns in our response, and we just wanted to kindly check whether anything remains unclear.
>
> If any part of our clarification could be improved, we would be very happy to provide further details.

---

### Official Review · Reviewer_6jPR · 2025-10-31

**Soundness:** 2
**Presentation:** 2
**Contribution:** 2
**Rating:** 4
**Confidence:** 4

**Summary:**

TTOM is a training-free framework for compositional text-to-video generation. It aligns model attention with LLM-generated spatiotemporal layouts via test-time optimization and stores optimized parameters in a parametric memory for reuse. This design enables continual adaptation across prompts, yielding significant improvements in motion, spatial alignment, and semantic consistency.

**Strengths:**

1. Clear motivation. the paper addresses a key limitation of video foundation models by improving their compositional understanding of objects, relations, attributes, and temporal dynamics within a realistic streaming generation setting.
2. Parametric memory flexibly manages prompt-level optimization results, enabling efficient transfer and adaptive refinement across prompts.

**Weaknesses:**

1. Line54 has two "in this paradigm”, typo error.
2. The cases shown in Figure 1, when compared with the baseline, reveal that the baseline also produces outputs that are partially semantically correct and sometimes exhibit higher visual quality — for instance, in the case of “An elderly man walking to the right in a sunny park.”
3. TTOM relies heavily on the layout quality produced by the LLM, assuming that the generated spatiotemporal layouts are accurate and consistent. However, layout hallucinations or spatial inaccuracies in the LLM outputs can directly cause alignment errors, misleading the optimization process and leading to degraded results.
4. There exists a structural trade-off between efficiency and quality.  Continuous TTO introduces significant latency, while memory mechanisms can reduce computational overhead but require a warm-up phase to accumulate sufficient samples.  Consequently, the deployment barrier remains high for online or low-latency applications such as real-time interactive editing.

**Questions:**

1. How strongly does TTOM depend on the LLM’s prompt design, model choice, or layout verification? And what happens when the LLM produces noisy or uncertain spatial temporal layouts, can TTOM still perform reliably?
2. Has the system been tested for cases where outdated or irrelevant memories build up and start to hurt performance? Are there situations where a faulty memory causes negative transfer? i hope the author can provide some comparison between successful and failure cases.
3. as shown in the above weakness part.

---

> ### Author Response · Authors · 2025-11-21
> **Author Response to Reviewer 6jPR (1/2)**
>
> We sincerely thank the reviewer for the thorough evaluation and insightful feedback. Your constructive comments have been invaluable in improving the clarity and overall quality of our manuscript. Our detailed point-by-point responses are provided below.
>
> ---
>
> > (W1) Line54 has two "in this paradigm”, typo error.
>
> Thanks for your comments. We have corrected this typo error.
>
> ---
> > (W2) The cases shown in Figure 1, when compared with the baseline, reveal that the baseline also produces outputs that are partially semantically correct and sometimes exhibit higher visual quality — for instance, in the case of “An elderly man walking to the right in a sunny park.”
>
> Thanks for raising this concern. According to Fig. 1 and the corresponding video files in the supplementary materials, we can see that even though the elderly man seems to move to the right in the 2D space, he is more likely to **walk forward toward the camera in the physical 3D world, which brings some ambiguity**. In contrast, the man generated by our method is evident to walk to the right.
>
> Besides, compared with the baseline result, the video generated by our method is more dynamic and realistic. For instance, **the smoothness, temporal consistency, and naturalness** of the elderly's movements are much better.
>
> ---
> > (W3) TTOM relies heavily on the layout quality produced by the LLM, assuming that the generated spatiotemporal layouts are accurate and consistent. However, layout hallucinations or spatial inaccuracies in the LLM outputs can directly cause alignment errors, misleading the optimization process and leading to degraded results.
>
> To make sure the trustworthiness of the proposed layout planning method via LLMs, we carried out a comprehensive human evaluation to layout planning accuracy across 7 categories of T2V-CompBench. In detail, we evaluate the original generated layouts and those after our proposed verification strategy, as shown below. These results verify that LLMs are reliable for performing spatial-temporal layout planning, and further verification could significantly improve the planning accuracy.
>
> | Verification | Success Rate for Layout Planning |
> |-------------|--------------|
> | -           | 87.1%        |
> | ✓           | **98.2%**        |
>
>
> ---
> > (W4) trade-off between efficiency and quality
>
> Thanks for raising this concern about the efficiency and quality trade-off. As shown in Tab. 4, in fact, our method could achieve a promising trade-off, with **a remarkable performance gain of 76% (0.2696 to 0.4754), with only an additional 0.5% latency (425s to 427s)**, by comparing the first two rows. Even though we perform continuous TTO, the larger 47.5% latency is also acceptable compared with the performance improvement.
>
> In real-world scenarios, we could deploy a much larger memory to cover a broad range of creation scenarios collected by real users during the cold start phase, and then flexibly decide whether to perform continuous TTO according to the preference for efficiency or quality. Compared with future large throughput requests, the cold start phrase for memory deployment seems much more marginal.
>
> ---
> > (Q1) How strongly does TTOM depend on the LLM’s prompt design, model choice, or layout verification? And what happens when the LLM produces noisy or uncertain spatial temporal layouts, can TTOM still perform reliably?
>
> Thanks for the insightful question. As shown in the tables below, we carried out a series of ablation studies on prompt design, model choice, and layout verification. The results demonstrate that these three elements could actually affect video generation by the layout planning process, but the performance perturbation seems acceptable, considering the strong layout planning abilities of current LLMs.
>
> | LLM                 | Prompt Design | Verification | Performance |
> |---------------------|----------------|--------------|-------------|
> | -                   | -              | -            | 31.64       |
> | gpt4o               | simple         | ×            | 38.62       |
> | gpt4o               | refined        | ×            | 47.86       |
> | gpt4o               | refined        | ✓            | **48.57**       |
> | Qwen3-Omni-Flash    | refined        | ✓            | 47.29       |
>
> To further verify the robustness of TTOM, we intentionally add noise to the planned layouts and feed them into our model for video generation, as shown below. The results demonstrate that TTOM could still perform reliably for noisy layouts, since the proposed learning objective in Eqn. (2) focuses on the global layout distribution alignment to resist noise interference.
> | Noise Type              | Performance |
> |-------------------------|-------------|
> | No noise                | 0.4857   |
> | Random perturbation     | 0.4821       |

---

> ### Author Response · Authors · 2025-11-21
> **Author Response to Reviewer 6jPR (2/2)**
>
> > (Q2) Has the system been tested for cases where outdated or irrelevant memories build up and start to hurt performance? Are there situations where a faulty memory causes negative transfer? i hope the author can provide some comparison between successful and failure cases.
>
> To study the impact of irrelevant memories, we conducted an experiment where we **intentionally loaded lower-ranked (less relevant) memory entries** to simulate mismatch scenarios. We compared loading the Rank-1 (best match), Rank-3, and Rank-5 (weaker/mismatched) entries against the baseline (Wan2.1-14B).
>
> As shown below, even when loading the Rank-5 entry-which represents a significantly weaker semantic match-the model achieves a score of 44.96, which is still dramatically superior to the baseline (31.64). The performance drop from Rank-1 (48.57) to Rank-5 (44.96) is gradual rather than catastrophic. This confirms that **TTOM is robust to memory mismatches** to some extent. The "generalizable motion priors" contained even in mismatched entries provide a better initialization. Predictably, a larger memory could provide a smoother semantic transition, with better robustness for missing the perfect match.
> | Method        | Performance |
> |---------------|-------------|
> | TTOM, Rank-1  | 48.57       |
> | TTOM, Rank-3  | 47.25       |
> | TTOM, Rank-5  | 44.96       |
> | Wan2.1-14B    | 31.64       |
>
>
> We also listed intuitive examples in **Fig. 8, in the Appendix**, to compare successful and failure cases. We can see that mismatches could hurt performance to some extent, but the partial semantic match is still helpful for better generation.

---

> ### Author Response · Authors · 2025-11-25
> **Follow-up on Discussion**
>
> Dear Reviewer 6jPR,
>
> Thank you once again for your thoughtful feedback and the time devoted to reviewing our submission. We have carefully addressed the concerns raised and provided detailed responses in this rebuttal.
>
> If any points remain unclear or require further clarification, we would be very happy to provide additional explanation or engage in further discussion.

---

> ### Author Response · Authors · 2025-11-26
> **Summary of User Response**
>
> Thank you again for your detailed and constructive feedback. We have carefully revised the manuscript and provided comprehensive responses to all your comments. In particular, we have:
>
> - **W1**: Corrected the duplicated phrase at Line 54.
>
> - **W2**: Clarified the semantic ambiguity in the baseline example and explained why TTOM produces more realistic motion and stronger temporal consistency.
>
> - **W3**: Provided human evaluation results demonstrating high reliability of LLM-based layout planning, and showed that our verification strategy significantly improves layout accuracy (87.1% → 98.2%).
>
> - **W4**: Expanded the discussion on the efficiency–quality trade-off, showing that TTOM achieves strong performance gains with marginal latency increases, and explaining practical deployment strategies with warm-start memory.
>
> - **Q1**: Added ablation studies on prompt design, model choice, and verification, as well as robustness experiments under intentionally noisy layouts.
>
> - **Q2**: Conducted mismatched-memory experiments (Rank-1/Rank-3/Rank-5) to evaluate negative transfer and demonstrated that TTOM degrades gracefully and remains robust even with irrelevant entries.
>
> We hope these revisions and additional analyses have fully addressed your concerns. If any points remain unclear or you would like further clarification, we would be very happy to provide more details.
>
> Thank you again for your time and thoughtful feedback.

---

> ### Author Response · Authors · 2025-11-27
>
> Dear Reviewer 6jPR,
>
> Thank you for your thoughtful comments and careful review. We have addressed each point in our rebuttal.
>
> If any part is still unclear, we’re glad to elaborate. We would be grateful if you could re-evaluate the paper considering these revisions.

---

### Official Review · Reviewer_9VTH · 2025-11-01

**Soundness:** 3
**Presentation:** 3
**Contribution:** 3
**Rating:** 4
**Confidence:** 3

**Summary:**

This paper proposes a framework named TTOM (Test-Time Optimization and Memorization) for improving compositional text-to-video (T2V) generation. During inference, TTOM aligns video generation results with spatiotemporal layouts and introduces lightweight, optimizable parameters for adaptive per-sample updates. It further incorporates a parameterized memory mechanism that supports insertion, retrieval, update, and deletion operations, enabling the reuse of historical optimization context. Experimental results on T2V-CompBench and VBench demonstrate substantial gains, especially in motion, quantity, and spatial relation metrics.

However, the paper suffers from a serious conceptual confusion. Although TTOM is repeatedly described as a “training-free” framework, the core algorithm explicitly performs gradient-based parameter updates (e.g., optimizing LoRA weights) during inference. Such optimization is, by definition, a form of test-time training or test-time adaptation, not a training-free operation. Therefore, labeling the method as “training-free” is conceptually incorrect and misleading. The authors must clarify this terminology to ensure methodological and scientific accuracy.

**Strengths:**

1. The paper presents an innovative test-time compositional generation framework equipped with a streaming memory mechanism, distinguishing it from conventional per-sample optimization approaches. This enables continual adaptation and historical context reuse during inference.

2. By unifying per-sample test-time optimization with a parametric memory module, the method achieves continual adaptation and efficient context reuse across inference streams, representing a novel direction in test-time generative modeling.

3. Extensive experiments provide strong empirical evidence of its effectiveness, showing performance improvements of +34.45% and +15.83% over CogVideoX-5B and Wan2.1-14B on T2V-CompBench (Table 1), together with notable gains in semantic consistency on VBench.

4. While the idea of “training-free test-time optimization” is conceptually appealing, the misuse of terminology undermines the paper’s scientific rigor. Any process involving gradient updates or parameter optimization cannot be described as training-free. The term must be corrected in the abstract, introduction, and method sections to maintain terminological precision.

**Weaknesses:**

**1. Conceptual ambiguity in the “training-free” claim:**

The paper repeatedly claims that TTOM is “training-free,” yet the method explicitly introduces learnable parameters and updates them via gradient descent during inference. Regardless of whether this occurs at test time, such an optimization process constitutes training—albeit without external training data. Hence, “training-free” and “test-time optimization” are not interchangeable concepts.
This is a critical conceptual flaw. The authors should clearly distinguish among:

- training-free: no gradient updates at all;

- fine-tuning-free: no updates to the backbone parameters, but auxiliary modules may be trained;

- test-time optimization: gradient-based updates performed at inference for adaptation.
Since TTOM belongs to the third category, it should be explicitly defined as a test-time optimization–based framework, not a training-free one.

**2. Lack of theoretical analysis of the memory mechanism:**

The insertion and update strategies in Equation (3) lack mathematical formalization and convergence analysis. The paper provides only an implementation-level description, without proving or empirically testing stability, information retention, or forgetting dynamics under continual updates.

**3. Dependence on LLM-generated layouts:**

Section 3.1 heavily relies on GPT-4o for layout generation but lacks robustness evaluation. The paper does not quantify how common layout errors (e.g., bounding-box misalignment, temporal inconsistencies) influence test-time optimization or how the “verification step” mitigates error propagation.

**4. Insufficient evaluation of computational and storage overhead:**

The latency and storage implications of the memory system (especially when full) are not systematically reported. The paper lacks detailed measurements of optimization time, memory growth (LoRA parameter accumulation), and replacement cost under streaming conditions.

**5. Lack of justification for hyperparameter choices:**

The paper sets LoRA rank=32 and optimizes only the first five denoising steps without explaining why these configurations are near-optimal. No sensitivity analysis is provided for LoRA rank, learning rate, or optimization steps, limiting generalization insight.

**Questions:**

**1. Clarification of “training-free” Definition**

The authors must explicitly state what “training-free” means in this context. Since the method involves gradient updates, how does it fundamentally differ from test-time training or test-time adaptation?

**2. Mathematical Modeling and Stability of Memory**

Could the memory insertion, update, and deletion processes be formalized (e.g., via state transition equations) and analyzed for stability or boundedness?

**3. Memory Capacity and Replacement Policies**

When memory reaches capacity, how do different replacement policies (LRU, LFU, FIFO) affect performance, latency, and storage footprint?

**4. Impact of Layout Errors on TTO**

How sensitive is TTOM to LLM-induced layout errors? How is the “verification step” implemented and evaluated quantitatively?

**5. Fault Tolerance and Error Accumulation**

How does TTOM prevent error propagation when mismatched memory entries are loaded or when the retrieved parameters are irrelevant?

**6. Fine-Grained Resource Analysis**

Could the authors provide detailed per-sample latency, GPU-hour costs, and LoRA parameter size during streaming?

**7. Hyperparameter Sensitivity**

Why were LoRA rank=32 and five denoising steps chosen? Are these settings robust across different backbones?

**8. Reproducibility and Randomness Control**

Please specify random seeds, GPT-4o prompt templates, and evaluation scripts for reproducibility.

**9. Failure Cases and Limitations**

What are the main failure modes (e.g., occlusion, excessive object count, ambiguous prompts)?

---

> ### Author Response · Authors · 2025-11-21
> **Author Response to Reviewer 9VTH (1/3)**
>
> We sincerely thank the reviewer for the thorough evaluation and insightful feedback. Your comments are highly constructive and have greatly helped us improve the clarity and quality of this manuscript. Our detailed, point-by-point responses are provided below.
>
> ---
>
> > (W1) Conceptual ambiguity in the "training-free" claim
>
> We thank the reviewer for this precise conceptual distinction. We agree that our method involves gradient updates and fits best under the definition of **Test-Time Optimization**. Our original use of "training-free" was intended to highlight that TTOM requires **no training phase with any external supervision**.
>
> To address this concern, we will revise the manuscript to explicitly define our method as a Test-Time Optimization (TTO) framework and modified the accoresponding descriptions which may bring confusion.
>
> ---
>
> > (W2) Lack of theoretical analysis of the memory mechanism
>
> Thanks for the comment. We have added comprehensive mathematical formalization and convergence analysis for Eqn. (3), including testing stability, information retention, or forgetting dynamics under continual updates, in **Sec. D of Appendix**.
>
> ---
> > (W3) Dependence on LLM-generated layouts
>
> Thanks for raising this concern. To delve into the influence of layout quality on video generation, we performed layout planning using a more lightweight LLM, i.e., Qwen3-Omni-Flash, and compared video generation results with those of GPT-4o, as shown below. It demonstrates that a stronger LLM would induce better video generation, due to the more accurate layout planning. Besides, the marginal performance gap also indicates that our method is robust to layout perturbation caused by weaker LLMs.
> | LLM                 | Prompt Design | Verification | Performance |
> |---------------------|----------------|--------------|-------------|
> | -                   | -              | -            | 31.64       |
> | gpt4o               | refined        | ✓            | **48.57**       |
> | Qwen3-Omni-Flash    | refined        | ✓            | 47.29       |
>
> Furthermore, we also carried out human evaluation on the layout planning performance to study the impact of the verification step. The results shown in the table below strongly demonstrates that 1) LLMs are very powerful to conduct spatial-temporal layout planning, and 2) the verification step could further mitagete possible planning errors.
>
> | Verification | Success Rate for Layout Planning |
> |-------------|--------------|
> | -           | 87.1%        |
> | ✓           | **98.2%**        |
>
> Finally, we also intentionally add noise to layouts and then perform generation with our model. The results verify the robustness of our method to noisy layouts.
> | Noise Type              | Performance |
> |-------------------------|-------------|
> | No noise                | 0.4857   |
> | Random perturbation     | 0.4821       |
>
> ---
> > (W4) Insufficient evaluation of computational and storage overhead
>
> We would like to clarify that the latency and optimization time have been already clearly reported in Tab. 4.
> The storage and memory growth is linear to the storage of LoRA parameters (105MB).
> Besides, the replacement cost mainly caused by retrieval. To quantify this, we conducted experiments measuring retrieval latency across varying memory sizes, as shown below.
>
> | Memory Size | Latency (ms) |
> |-------------|--------------|
> | 100         | 0.077        |
> | 1,000       | 0.082        |
> | 10,000      | 0.095        |
> | 100,000     | 0.472        |
>
> The results demonstrate that retrieval remains highly efficient even with large memory sizes, **incurring only 0.472 ms latency** at a scale of 100,000 entries.
>
>
> ---
> > (W5) Lack of justification for hyperparameter choices
>
> We would like to clarify that we have already conducted a detailed sensitivity analysis of the two most important hyperparameters, i.e., the denoising steps and optimization steps, **in Tab. 7 and Tab. 8**, respectively. Other settings, including LoRA rank and learning rate, were not included originally considering their relative insignificance.
>
> Now we add them in the tables below, which demonstrates that our method is robust to these hyperparameter settings within a proper range.
>
> | LoRA Rank | Performance |
> |-----------|-------------|
> | 16        | 45.31       |
> | 32        | 48.57       |
> | 64        | 48.49       |
> | 128       | 46.66       |
>
> | Learning Rate | Performance |
> |---------------|-------------|
> | 1e-3          | 45.82       |
> | 5e-3          | 46.94       |
> | 1e-4          | 48.57       |
> | 5e-5          | 47.22       |
>
> ---
> > (Q1) Clarification of "training-free" Definition
>
> As discussed in W1, we have revised this definition to make it clearer.
>
> ---
> > (Q2) Mathematical Modeling and Stability of Memory
>
> Please see the response to W2.

---

> ### Author Response · Authors · 2025-11-21
> **Author Response to Reviewer 9VTH (2/3)**
>
> > (Q3) Memory Capacity and Replacement Policies
>
> We thank the reviewer for this insightful question regarding memory management in streaming scenarios. To address this, we conducted an additional ablation study comparing LRU, LFU, and FIFO policies. We evaluated these policies on the T2V-CompBench Motion category (streaming setting) with a fixed memory capacity of 8 items and a total number of streaming prompts of 40. As shown in the table below.
>
> We give the following in-depth analysis: FIFO performs the worst because it discards entries blindly, often removing useful motion cues early. LRU improves upon this by keeping recently accessed items, which aligns better with temporal locality in streaming inputs. LFU achieves the best performance, as frequently referenced motion cues are consistently preserved while redundant ones are replaced.
>
> | Policy | Performance |
> |--------|-------------|
> | FIFO   | 48.53       |
> | LRU    | 51.64       |
> | LFU    | **52.74**       |
>
>
> **Impact on Performance**. LFU yields the highest performance, since it retains those high-utility "foundation" memory, ensuring they are available for knowledge transfer. In contrast, FIFO performs worst as it indiscriminately discards optimized parameters regardless of their utility or recency, forcing the model to relearn common patterns from scratch.
>
> **Impact on Latency**. Latency in TTOM is inversely related to the Memory Hit Rate. When a relevant layout is found in memory (Hit), the model loads parameters and skips or shortens optimization. When a miss occurs, the model must perform full TTO, which incurs higher latency. LFU maximizes the hit rate for common compositional queries, thereby minimizing the average inference latency compared to FIFO.
>
> **Impact on Storage Footprint**. The storage footprint is determined by the memory capacity (set to $N$ items) rather than the replacement policy. The memory stores LoRA parameters ($\phi^*$). The metadata overhead for tracking frequency (LFU) or timestamps (LRU) is negligible compared to the parameter weights (approx. 105MB per item with rank 32). Thus, the footprint remains constant across policies given a fixed capacity.
>
> ---
> > (Q4) Impact of Layout Errors on TTO
>
> Please see the response to W3.
>
> ---
> > (Q5) Fault Tolerance and Error Accumulation
>
> To quantitatively verify fault tolerance, we conducted an experiment where we **intentionally loaded lower-ranked (less relevant) memory entries to simulate mismatch scenarios**. We compared loading the Rank-1 (best match), Rank-3, and Rank-5 (weaker/mismatched) entries against the baseline (Wan2.1-14B).
>
> As shown below, even when loading the Rank-5 entry-which represents a significantly weaker semantic match-the model achieves a score of 44.96, which is still dramatically superior to the baseline (31.64). The performance drop from Rank-1 (48.57) to Rank-5 (44.96) is gradual rather than catastrophic. This confirms that **TTOM is robust to memory mismatches** to some extent. The "generalizable motion priors" contained even in mismatched entries provide a better initialization than random initialization, and the subsequent TTO step effectively realigns them to the correct context.
> | Method        | Performance |
> |---------------|-------------|
> | TTOM, Rank-1  | 48.57       |
> | TTOM, Rank-3  | 47.25       |
> | TTOM, Rank-5  | 44.96       |
> | Wan2.1-14B    | 31.64       |
>
> ---
> > (Q6) Fine-Grained Resource Analysis
>
> We have already provided detailed per-sample latency in **Tab. 4**. The GPU-hour costs is theoretically linear to the processed samples during stream. The per-sample LoRA parameter size (52.4M with 32 ranks) is invariant and the total size depends on the memory capacity linearly, which are irrelevant to the streaming setting.
>
> ---
> > (Q7) Hyperparameter Sensitivity
>
> As a common strategy, the hyperparameter sensitivity for LoRA rank=32 and 5 denoising steps was determined empirically, as shown in the tables below and Tab. 7, respectively.
>
> To verify the robustness to different backbones, we also carried out experiments with a different backbone, i.e., CogVideoX. The results shown below veirfy the robustness of the proposed method across different backbones.
>
> | LoRA Rank | Wan2.1 | CogVideoX |
> |-----------|--------|-----------|
> | 16        | 0.4531 | 0.4053    |
> | 32        | 0.4857 | 0.4409    |
> | 64        | **0.4849** | **0.4387**    |
> | 128       | 0.4666 | 0.4302    |
>
> | Timestep | Wan2.1 | CogVideoX |
> |----------|--------|-----------|
> | 1        | 0.2730 | 0.2689    |
> | 3        | 0.3692 | 0.3507    |
> | 5        | **0.4321** | **0.4302**    |
> | 7        | 0.4296 | 0.4027    |
>
> | Learning Rate | Wan2.1 | CogVideoX |
> |---------------|--------|-----------|
> | 1e-3          | 0.4582 | 0.4049    |
> | 5e-3          | 0.4694 | 0.4214    |
> | 1e-4          | **0.4857** | **0.4302**    |
> | 5e-5          | 0.4722 | 0.4283    |

---

> ### Author Response · Authors · 2025-11-21
> **Author Response to Reviewer 9VTH (3/3)**
>
> > (Q8) Reproducibility and Randomness Control
>
> To ensure reproducibility, we set the random seed as 42, which has already been specified in the attached code.
> We have added the GPT-4o prompt templates in Appendix XXX.
>
> As for the evaluation, we follow the official scripts released in the GitHub repositories of T2V-CompBench and VBench.
>
> ---
> > (Q9) Failure Cases and Limitations
>
> We show some **failure cases in Fig. 10, Appendix**.
>
> - The first example shows that dealing with excessive object count is still challenging for current video generative models, even though our method generates a more accurate count than the baseline. One of the reasons is that the model lacks sufficient physical law modeling abilities, so it is weak in solving the occlusion problem.
> - In the second example, the layout planned by LLM is correct, but video generation fails due to the relative motion of the object and the camera. We leave this challenging direction for camera motion as future work.

---

> ### Author Response · Authors · 2025-11-25
> **Follow-up on Discussion**
>
> Dear Reviewer 9VTH,
>
> Thank you once again for your thoughtful feedback and the time devoted to reviewing our submission. We have carefully addressed the concerns raised and provided detailed responses in this rebuttal.
>
> If any points remain unclear or require further clarification, we would be very happy to provide additional explanation or engage in further discussion.

---

> ### Author Response · Authors · 2025-11-26
> **Summary of User Response**
>
> Thank you again for your detailed and constructive feedback. We have carefully revised the manuscript and provided comprehensive responses to all your comments. In particular, we have:
>
> - **W1 / Q1**: Clarified the conceptual terminology and explicitly defined TTOM as a test-time optimization framework rather than “training-free.”
>
> - **W2 / Q2**: Added a mathematical formalization of the memory mechanism, including stability and update dynamics, in the Appendix.
>
> - **W3 / Q4**: Conducted robustness studies on layout quality, evaluated the verification step, and compared different LLM planners.
>
> - **W4 / Q6**: Expanded the analysis of computational cost, retrieval latency, and storage growth under streaming settings.
>
> - **W5 / Q7**: Included new sensitivity analyses for LoRA rank, learning rate, and timesteps across two backbone models.
>
> - **Q3**: Added ablation results and analysis for replacement policies (FIFO/LRU/LFU).
>
> - **Q5**: Verified fault tolerance through mismatched-memory experiments.
>
> - **Q8**: Added random seeds, GPT-4o prompt templates, and evaluation details for reproducibility.
>
> - **Q9**: Included additional failure cases and discussed limitations.
>
> We hope these revisions have fully addressed your concerns. If any points remain unclear or require further clarification, we would be very happy to provide additional details.
>
> Thank you again for your time and thoughtful feedback.

---

> > ### Comment · Reviewer_9VTH · 2025-11-28
> >
> > I appreciate the authors’ detailed responses and the additional clarifications provided. The rebuttal addresses several of my earlier questions, and the newly added explanations help improve the overall readability of the submission. Below I provide several follow-up comments and suggestions.
> >
> > ---
> >
> > **Clarity of the Streaming Setting.**
> >
> > The authors clarified the pseudo-training procedure used for constructing the memory before evaluation. While the explanation is helpful, I still believe the manuscript would benefit from a more explicit and consolidated description of this setting in the main paper — especially regarding (1) the motivation for separating pseudo-training prompts from test prompts, and (2) how this affects reproducibility.
> >
> > ---
> >
> > **Scope and Limitations of the Memory Mechanism.**
> >
> > The memory mechanism is a central component of the method. Although the authors added additional justification, some limitations remain underexplored. For example, how memory quality degrades when the prompt distribution shifts significantly, or how retrieval behaves when multiple partially related entries exist. I do not expect new experiments, but a clearer discussion section would improve transparency.
> >
> > ---
> >
> > **Dependency on LLM-Generated Layouts.**
> >
> > The authors confirmed that layout quality from LLMs is critical. It may still be useful to include a brief note describing typical failure cases (e.g., incorrect temporal ordering, misinterpreted spatial relations) and how sensitive the TTO process is to such inaccuracies.
> >
> > ---
> >
> > **Complexity & Practicality.**
> > The efficiency analysis is appreciated, but I still encourage the authors to explicitly state the computational overhead of maintaining and querying the parametric memory in real-world deployment, even if approximate.
> >
> > Overall, the authors’ responses were constructive and improved my understanding of the method

---

> > > ### Author Response · Authors · 2025-11-29
> > >
> > > Thank you for your continued feedback and for acknowledging our efforts to address your concerns. We respond to the new questions below in turn.
> > >
> > > ---
> > > 1. Clarity of the Streaming Setting.
> > >
> > > In fact, we have provided a detailed explanation of the motivation and implementation of the *test-time independence setting* in Streaming Video Generation Setting in **Sec. 4.1 of the original manuscript**:
> > > > To align with practical video creation scenarios where user prompts arrive sequentially, we propose modeling historical streaming contexts. For a fair comparison with prior methods that treat test samples independently, we introduce a *test-time independence* setting. Specifically, we generate 200 prompts covering common compositional scenes as *pseudo-training data*, via GPT-4o, and subsequently generate videos from these prompts while applying TTOM to construct a memory.
> > >
> > >
> > > The key motivation is to **ensure a fair comparison** with prior methods that treat test samples independently. This setting has **no impact on reproducibility**, as full implementation details and source code are provided to ensure complete transparency.
> > >
> > > ---
> > > 2. Scope and Limitations of the Memory Mechanism.
> > >
> > > Our memory mechanism is designed to be **robust to prompt distribution shifts**. When a prompt is out-of-distribution (OOD), the similarity between the prompt and all memory entries is low. In such cases, we default to applying TTO without memory, subsequently incorporating the optimized result into memory—allowing adaptation to distributional drift over time.
> > >
> > > Regarding partially related entries, we have addressed this explicitly in **our response to `Q5`**, supported by empirical results demonstrating the method’s robustness in such scenarios.
> > >
> > > ---
> > > 3. Dependency on LLM-Generated Layouts.
> > >
> > > As clarified in **the response to `W3`**, we have pointed out that layout planning could be improved to a 98.2% success rate with the proposed verification step. We do not observe failure cases like incorrect temporal ordering or misinterpreted spatial relations. **A representative failture case without verification** is that LLMs sometimes output a sinlge bounding box (bbox) labeled "five pandas" rather than 5 distinct bboxes for individual pandas, which may affect the subsequent video generation with TTO. We will include this discussion in the final version to highlight the effectiveness of the verification step.
> > >
> > > Besides, **our noise injection experiments in the response to `W3`** further demonstrate the robustness of TTO to layout imperfections.
> > >
> > > ---
> > > 4. Complexity & Practicality.
> > >
> > > In fact, as detailed in **the response to `W4`**, we have explicitely stated the computational overhead under a wide range of scales, from 100 to 100,000, simulating real-world deployment scenarios. Even at 100,000 entries, the added latency is minimal—only 0.472 ms. While testing at larger scales is limited by our computational resources, we believe our experiments are sufficient to validate the memory mechanism’s efficiency under practical constraints.
> > >
> > > **In summary, all four concerns have been discussed thoroughly in our previous responses**, with supporting experiments and references provided for your convenience. We appreciate your time and consideration in re-evaluating our submission.

---

> ### Author Response · Authors · 2025-11-27
>
> Dear Reviewer 9VTH,
>
> Thank you for your thoughtful comments and careful review. We have addressed each point in our rebuttal.
>
> If any part is still unclear, we’re glad to elaborate. We would be grateful if you could re-evaluate the paper considering these revisions.

---

### Official Review · Reviewer_f1wz · 2025-11-02

**Soundness:** 3
**Presentation:** 3
**Contribution:** 3
**Rating:** 6
**Confidence:** 4

**Summary:**

This paper introduces **TTOM**, a training-free framework that enhances compositional text-to-video (T2V) generation through *test-time optimization (TTO)* and a *parametric memory* mechanism. TTOM aligns cross-attention maps of video foundation models (VFMs) with LLM-generated spatiotemporal layouts at inference time, improving text–video compositional alignment. Its parametric memory stores optimized parameters from previous prompts, enabling reuse, continual learning, and efficient streaming inference. Experiments on **T2V-CompBench** and **VBench** demonstrate significant improvements in compositional reasoning, semantic consistency, and visual quality over baselines such as CogVideoX‑5B and Wan2.1‑14B.

**Strengths:**

- Proposes a novel test-time optimization framework with memory-based reuse, extending inference-time adaptation for video generation in a principled way.
- Seamlessly integrates LLM-driven spatiotemporal layout planning for controllable, interpretable video synthesis.
- Achieves consistent quantitative and qualitative gains on major T2V benchmarks, including clear improvements in motion and numeracy.
- The design supports streaming inference and user-specific memory, aligning well with practical workflows.
- Comprehensive ablations, detailed comparisons, and visual evaluations substantiate the method’s effectiveness.

**Weaknesses:**

1. The method is largely empirical; there is a lack of theoretical explanation or analysis regarding optimization stability and memory dynamics over time.
2. Evaluation scenarios mainly focus on compositional prompts; testing under more complex, long-horizon, or open-domain conditions would strengthen generalization claims.
3. Dependence on large LLMs (e.g., GPT‑4o) for layout planning introduces computational and cost overheads that are not thoroughly discussed.
4. Runtime performance and latency analysis are insufficient; the paper should quantify the cost of test-time optimization steps and memory operations.
5. Some ablations, while numerically helpful, could benefit from visual examples or error diagnostics to clarify how memory and optimization complement each other.

**Questions:**

see the weakness

---

> ### Author Response · Authors · 2025-11-21
> **Author Response to Reviewer f1wz**
>
> We are deeply grateful to the reviewer for the thorough evaluation and thoughtful feedback. Your comments are highly constructive and have been instrumental in helping us refine this manuscript. Our point-by-point responses are provided below.
>
> ---
>
> > (W1) the method is largely empirical; there is a lack of theoretical explanation or analysis regarding optimization stability and memory dynamics over time.
>
> Thanks for your suggestion. We have added a comprehensive theoretical explanation in Sec. D of Appendix.
>
>
> ---
>
> > (W2) Evaluation scenarios mainly focus on compositional prompts; testing under more complex, long-horizon, or open-domain conditions would strengthen generalization claims.
>
> Thanks for your constructive advice. Compositional video generation has been a challenging task, and even state-of-the-art T2V models [a] are still struggling with it [b]. Performance improvement on compositional benchmarks could demonstrate the progress of video generative models. T2V-CompBench and VBench used in our work are the two most commonly used benchmarks. Besides, more efforts need to be made to contribute to more challenging benchmarks containing more complex, long-horizon, or open-domain conditions in the future.
>
> Considering the unavailability of more challenging benchmarks currently, we **collected more complex prompts and performed human evaluation** from two aspects (Prompt Following and Visual Quality) to evaluate our method, as shown in the table below. We also list the qualitative results in **Fig. 9 in Appendix**. The results (especially the **significant improvement on Prompt Following**) could further verify the effectiveness, superiority, and generalization of our method.
>
> | Model | Prompt Following | Visual Quality |
> |-------|-------------------|----------------|
> | Wan2.1-14B (Baseline)   | 0.28              | 0.71           |
> | ours  | **0.67**              | **0.76**           |
>
> [a] Wan: Open and Advanced Large-Scale Video Generative Models. arXiv'25
>
> [b] T2V-CompBench: A Comprehensive Benchmark for Compositional Text-to-video Generation. arXiv'24
>
> ---
> > (W3) Dependence on large LLMs (e.g., GPT-4o) for layout planning introduces computational and cost overheads that are not thoroughly discussed.
>
> Thanks for raising this concern. Based on the evaluation of the computational and cost overheads, we observed that generating a layout for GPT-4o consumes about **1,539 input tokens and 1,059 output tokens, spending 3s**, on average. It demonstrates that the layout planning process for LLMs is *very efficient and computationally friendly*.
>
> ---
> > (W4) Runtime performance and latency analysis are insufficient; the paper should quantify the cost of test-time optimization steps and memory operations.
>
> We first show the runtime latency with different test-time optimization steps in the table below. Besides, Tab. 7 and Tab. 8 demonstrate that we only need fewer than 10 optimization steps and 5 denoising timesteps to achieve significant performance gain in practice. Specifically, we could get **14% performance improvement (0.3789 to 0.4321)** with **an additional runtime latency of less than 1.8%**, indicating a promising trade-off.
> | Steps of TTO | Latency (s) |
> |--------------|-------------|
> | 0            | 425         |
> | 10           | 503         |
> | 20           | 580         |
> | 30           | 661         |
>
> We provide a breakdown of the runtime cost for each memory operation below:
> - Insert: ~0s (dominated by minimal I/O overhead)
> - Read: 0.1867 s
> - Write: 0.2235 s
> - Update: ~0s (mainly for I/O overhead)
>
> The above results demonstrate that the runtime cost of the memory mechanism is almost negligible, **only accounting for 0.08%** (0.4102s vs. 503s).
>
> ---
> (W5) Some ablations, while numerically helpful, could benefit from visual examples or error diagnostics to clarify how memory and optimization complement each other.
>
> Thanks for your comments. We add more visual examples in **Fig. 7 in Appendix** to show the complementary relations between memory and optimization. For these examples, we can intuitively see the effectiveness of the proposed test-time optimization and memory mechanism.

---

> ### Author Response · Authors · 2025-11-25
> **Follow-up on Discussion**
>
> Dear Reviewer f1wz,
>
> Thank you again for your time and for the positive evaluation. We truly appreciate your careful review. We have addressed the concerns in our response, and we just wanted to kindly check whether anything remains unclear.
>
> If any part of our clarification could be improved, we would be very happy to provide further details.

---

> ### Author Response · Authors · 2025-11-26
> **Summary of User Response**
>
> Thank you again for your detailed and constructive feedback. We have carefully revised the manuscript and provided comprehensive responses to all your comments. In particular, we have:
>
> - **W1**: Added a theoretical explanation of optimization stability and memory dynamics in Appendix Sec. D.
> - **W2**: Extended evaluation to more complex prompts through human studies (Prompt Following & Visual Quality) and added qualitative results in Fig. 9.
> - **W3**: Provided computational and cost analysis of LLM-based layout planning, showing that GPT-4o layout generation is efficient.
> - **W4**: Added detailed runtime and latency analysis for TTO steps and memory operations, demonstrating minimal additional overhead.
> - **W5**: Added visual examples (Fig. 7) illustrating how memory and test-time optimization complement each other.
>
> We hope these revisions have fully addressed your concerns. If any points remain unclear or require further clarification, we would be very happy to provide additional details.
>
> Thank you again for your time and thoughtful feedback.

---

> ### Author Response · Authors · 2025-11-27
>
> Dear Reviewer f1wz,
>
> Thank you for your thoughtful comments and careful review. We have addressed each point in our rebuttal.
>
> If any part is still unclear, we’re glad to elaborate. We would be grateful if you could re-evaluate the paper considering these revisions.

---

### Meta-Review · Area_Chair_prRt · 2026-01-03

**Summary:**

There are mixed reviews for paper 3050 with 2 accept (8 and 6) and 2 weakly reject.


During the review process, here are the main concerns from the reviewers:
1. Over-claim of training-free since the proposed method need test-time optimization.
2. How strong the correlation between LLM prompt design, model choice, or layout verification with the final results
3. Inference Efficiency and Latency Overhead

In general, authors do provide a comprehensive rebuttal for the major problems that authors provided. But I am concerned on the fact that these are important parts that should be in original paper but not in the original paper.

**Reviewer Concerns:**

From the rebuttal, I do see that all the major concerns are addressed.

**Reviewer Scores:**

There are mixed reviews for paper 3050 with 2 accept (8 and 6) and 2 weakly reject.

---

### Decision · Program_Chairs · 2026-01-26

Accept (Poster)